# SELF-SUPERVISED GAN COMPRESSION

## ABSTRACT

Deep learning's success has led to larger and larger models to handle more and more complex tasks; trained models can contain millions of parameters. These large models are compute- and memory-intensive, which makes it a challenge to deploy them with minimized latency, throughput, and storage requirements. Some model compression methods have been successfully applied on image classification and detection or language models, but there has been very little work compressing generative adversarial networks (GANs) performing complex tasks. In this paper, we show that a standard model compression technique, weight pruning, cannot be applied to GANs using existing methods. We then develop a self-supervised compression technique which uses the trained discriminator to supervise the training of a compressed generator. We show that this framework has a compelling performance to high degrees of sparsity, can be easily applied to new tasks and models, and enables meaningful comparisons between different pruning granularities.

## 1 INTRODUCTION

Deep Neural Networks (DNNs) have proved successful in various tasks like computer vision, natural language processing, recommendation systems, and autonomous driving. Modern networks are comprised of millions of parameters, requiring significant storage and computational effort. Though accelerators such as GPUs make realtime performance more accessible, compressing networks for faster inference and simpler deployment is an active area of research. Compression techniques have been applied to many networks, reducing memory requirements and improving their performance. Though these approaches do not always harm accuracy, aggressive compression can adversely affect the behavior of the network. Distillation (Schmidhuber, 1991; Hinton et al., 2015) can improve the accuracy of a compressed network by using information from the original, uncompressed network.

Generative Adversarial Networks (GANs) (Schmidhuber, 1990; Goodfellow et al., 2014) are a class of DNN that consist of two sub-networks: a generative model and a discriminative model. Their training process aims to achieve a Nash Equilibrium between these two sub-models. GANs have been used in semi-supervised and unsupervised learning areas, such as fake dataset synthesis (Radford et al., 2016; Brock et al., 2019), style transfer (Zhu et al., 2017b; Azadi et al., 2018), and image-to-image translation (Zhu et al., 2017a; Choi et al., 2018). As with networks used in other tasks, GANs have millions of parameters and nontrivial computational requirements.

In this work, we explore compressing the generative model of GANs for more efficient deployment. We show that applying standard pruning techniques, with and without distillation, can cause the generator's behavior to no longer achieve the network's goal. Similarly, past work targeted at compressing GANs for simple image synthesis fall short when they are applied to large tasks. In some cases, this result is masked by loss curves that look identical to the original training. By modifying the loss function with a novel combination of the pre-trained discriminator and the original and compressed generators, we can overcome this behavioral degradation and achieve compelling compression rates with little change in the quality of the compressed generator's ouput. We apply our technique to several networks and tasks to show generality. Finally, we study the behavior of compressed generators when pruned with different amounts and types of sparsity, finding that filter pruning, a technique commonly used for accelerating image classification networks, is not trivially applicable to GANs.

Our main contributions are:

- We illustrate that and explain why pruning the generator of a GAN with existing methods is unsatisfactory for complex tasks. (Section 3)
- We propose self-supervised compression for the generator in a GAN (Section 4)
- We show that our technique can apply to several networks and tasks (Section 5)
- We show and analyze qualitative differences in pruning ratio and granularities. (Section 6)

## 2 RELATED RESEARCH

A common method of DNN compression is network pruning (Han et al., 2015): setting the small weights of a trained network to zero and fine-tuning the remaining weights to recover accuracy. Zhu & Gupta (2018) proposed a gradual pruning technique (AGP) to compress the model during the initial training process. Wen et al. (2016) proposed a structured sparsity learning method that uses group regularization to force weights towards zero, leading to pruning groups of weights together. Li et al. (2017) pruned entire filters and their connecting feature maps from models, allowing the network to run with standard dense software libraries. Though it was initially applied to image classification networks, network pruning has been extended to natural language processing tasks (See et al., 2016; Narang et al., 2017) and to recurrent neural networks (RNNs) of all types - vanilla RNNs, GRUs (Cho et al., 2014), and LSTMs (Hochreiter & Schmidhuber, 1997). As with classification networks, structured sparsity within recurrent units has been exploited (Wen et al., 2018).

A complementary method of network compression is quantization. Sharing weight values among a collection of similar weights by hashing (Chen et al., 2015) or clustering (Han et al., 2016) can save storage and bandwidth at runtime. Changing fundamental data types adds the ability to accelerate the arithmetic operations, both in training (Micikevicius et al., 2018) and inference regimes (Jain et al., 2019).

Several techniques have been devised to combat lost accuracy due to compression, since there is always the chance that the behavior of the network may change in undesirable ways when the network is compressed. Using GANs to generate unique training data (Liu et al., 2018b) and extracting knowledge from an uncompressed network, known as distillation (Hinton et al., 2015), can help keep accuracy high. Since the pruning process involves many hyperparameters, Lin et al. (2019) use a GAN to guide pruning, and Wang et al. (2019a) structure compression as a reinforcement learning problem; both remove some of the burden from the user.

## 3 EXISTING TECHNIQUES FAIL TO PRUNE A COMPLEX TASK

Though there are two networks in a single GAN, the main workload at deployment is usually from the generative model, or generator. For example, in image synthesis and style transfer tasks, the final output images are created solely by the generator. The discriminative model (discriminator) is vital in training, but it is abandoned afterward for many tasks. So, when we try to apply state-of-the-art compression methods to GANs, we focus on the generator for efficient deployment. As we will see, the generative performance of the compressed generators is quite poor for the selected image-to-image translation task. We look at two broad categories of baseline approaches: standard pruning techniques that have been applied to other network architectures, and techniques that were devised to compress the generator of a GAN performing image synthesis. We compare to the dense baseline [a], our technique [b], as well as a small, dense network with the same number of parameters [c]. (Labels correspond to entries in Table 1, the overview of all techniques, and Figure 1, results of each technique).

**Standard Pruning Techniques**. To motivate GAN-specific compression methods, we try variations of two state-of-the-art pruning methods: manually pruning and fine tuning (Han et al., 2015) a trained dense model [d], and AGP (Zhu & Gupta, 2018) from scratch [e] and during fine-tuning [f]. We also include distillation (Hinton et al., 2015) to improve the performance of the pruned network with manual pruning [g] and AGP fine-tuning [h]. Distillation is typically optional for other network types, since it is possible to get decent accuracy with moderate pruning in isolation. For very aggressive compression or challenging tasks, distillation aims to extract knowledge for the compressed (student) network from original (teacher) network's behavior. We also fix the discriminator of [g] to see if the discriminator was being weakened by the compressed generator [i].

**Targeted GAN Compression**. There has been some work in compressing GANs with methods other than pruning, and only one technique applied to an image-to-image translation task. We first examine two approaches similar to ours. Adversarial training (Wang et al., 2018) [j] posits that during distillation of a classification network, the student network can be thought of as a generative model attempting to produce features similar to that of the teacher model. So, a discriminator was trained alongside the student network, trying to distinguish between the student and the teacher. One could apply this technique to compress the generator of a GAN, but we find that its key shortcoming is that it trains a discriminator from scratch. Similarly, distillation has been used to compress GANs in Aguinaldo et al. (2019) [k], but again, the "teacher" discriminator was not used when teaching the "student" generator.

Learned Intermediate Representation Training (LIT) (Koratana et al., 2019) [l] compresses StarGAN by a factor of $1.8\times$ by training a shallower network. Crucially, LIT does not use the pre-trained discriminator in any loss function. Quantized GANs (QGAN) (Wang et al., 2019b) [m] use a training process based on Expectation-Maximization to achieve impressive compression results on small generative tasks with output images of 32x32 or 64x64 pixels. Liu et al. (2018a) find that maintaining a balance between discriminator and generator is key: their approach is to selectively binarize parts of both networks in the training process on the Celeb-A generative task, up to 64x64 pixels. So, we try pruning both networks during the training process [n].

**Experiments**. For these experiments, we use StarGAN (Choi et al., 2018) trained with the Distiller (Zmora et al., 2018) library for the pruning. StarGAN[1] extends the image-to-image translation capability from two domains to multiple domains within a single unified model. It uses the Celeb-Faces Attributes (CelebA) (Liu et al., 2015) as the dataset. CelebA contains 202,599 images of celebrities' faces, each annotated with 40 binary attributes. As in the original work, we crop the initial images from size $178 \times 218$ to $178 \times 178$, then resize them to $128 \times 128$ and randomly select 2,000 images as the test dataset and use remaining images for training. The aim of StarGAN is facial attribute translation: given some image of a face, it generates new images with five domain attributes changed: 3 different hair colors (black, blond, brown), different gender (male/female), and different age (young/old). Our target sparsity is 50% for each approach.

We stress that we attempted to find good hyperparameters when using the existing techniques, but standard approaches like reducing the learning rate for fine-tuning (Han et al., 2015), etc., were not helpful. Further, the target sparsity, 50%, is not overly aggressive, and we do not impose any structure; other tasks readily achieve 80%-90% fine-grained sparsity with minimal accuracy impact.

The results of these trials are shown in Figure 1. Subjectively, it is easy to see that the existing approaches (1c through 1n) produce inferior results to the original, dense generator. Translated facial images from pruning & naïve fine-tuning (1d and 1e) do give unique results for each latent variable, but the images are hardly recognizable as faces. These fine-tuning procedures, along with AGP from scratch (1f) and distillation from intermediate representations (1l), simply did not converge. One-shot pruning and traditional distillation (1g), adversarial learning (1j), knowledge distillation (1k), training a "smaller, dense" half-sized network from scratch (1c) and pruning both generator and discriminator (1n) keep facial features intact, but the image-to-image translation effects are lost to mode collapse (see below). There are obvious mosaic textures and color distortion on the translated images from fine-tuning & distillation (1h), without fine-tuning the original loss (1i), and from the pruned model based on the Expectation-Maximization (E-M) algorithm (1m). However, the translated facial images from a generator compressed with our proposed self-supervised GAN compression method (1b) are more natural, nearly indistinguishable from the dense baseline (1a), matching the quantitative Frechet Inception Distance (FID) scores (Heusel et al., 2017) in Table 1. While past approaches have worked to prune some networks on other tasks (DCGAN generating MNIST digits, see the supplementary material), we show that they do not succeed on larger image-to-image translation tasks, while our approach works on both. Similarly, though LIT (Koratana et al., 2019) [l] was able to achieve a compression rate of $1.8\times$ on this task by training a shallower network, it does not see the same success at network pruning.

**Discussion**. It is tempting to think that the loss curves of the experiment for each technique can tell us if the result is good or not. We found that for many of these experiments, the loss curves correctly predicted that the final result would be poor. However, the curves for [h] and [m] look very

---

[1]StarGAN baseline implementation: `https://github.com/yunjey/StarGAN`.

Table 1: GAN compression comparison (network pruning)

| Technique | Generator(s) Compressed | Generator(s) Init Scheme | Discriminator Init Scheme | Discriminator Fixed | L-Gc | L-Dc | L-Go | L-Do | Results Qualitative | Results FID Score |
|---|---|---|---|---|---|---|---|---|---|---|
| (a) No Compression | Dense | Random | Dense,Random | No | - | - | Yes | Yes | Good | 6.113 |
| (b) Self-Supervised **(ours)** | Dense,Sparse | From Dense | Dense,Pretrained | No | Yes | Yes | Yes | Yes | Good | 6.929 |
| (c) Small & Dense Network | Dense | Random | Dense,Random | No | - | - | Yes | Yes | Mode collapse | 72.821 |
| (d) One-shot Pruning & Fine-Tuning | Sparse | From Dense | Dense,Pretrained | No | Yes | Yes | - | - | Facial artifacts | 24.404 |
| (e) Gradual Pruning & Fine-Tuning | Sparse | From Dense | Dense,Random | No | Yes | Yes | - | - | Facial artifacts | 35.677 |
| (f) Gradual Pruning during Training | Sparse | Random | Dense,Random | No | Yes | Yes | - | - | No faces | 84.941 |
| (g) One-shot Pruning & Distillation | Dense,Sparse | From Dense | - | - | Yes | - | Yes | - | Mode collapse | 45.461 |
| (h) (d) & Distillation | Dense,Sparse | From Dense | Dense,Pretrained | No | Yes | Yes | Yes | - | Color artifacts | 38.985 |
| (i) (g) & Fix Original Loss | Dense,Sparse | From Dense | Dense,Pretrained | Yes | Yes | Yes | - | - | Facial artifacts | 15.182 |
| (j) Adversarial Learning | Dense,Sparse | Random | Dense,Random | No | Yes | Yes | Yes | Yes | Mode collapse | 92.721 |
| (k) Knowledge Distillation | Dense,Sparse | From Dense | Dense,Random | No | Yes | - | Yes | Yes | Mode collapse | 103.094 |
| (l) Distill Intermediate (LIT) | Dense,Sparse | From Dense | Dense,Pretrained | Yes | - | - | - | - | No faces | 194.026 |
| (m) E-M Pruning | Dense,Sparse | From Dense | Sparse,Pretrained | No | Yes | Yes | Yes | - | Color artifacts | 159.767 |
| (n) G & D Both Pruning | Dense,Sparse | From Dense | Sparse,Pretrained | No | Yes | Yes | Yes | - | Mode collapse | 46.453 |

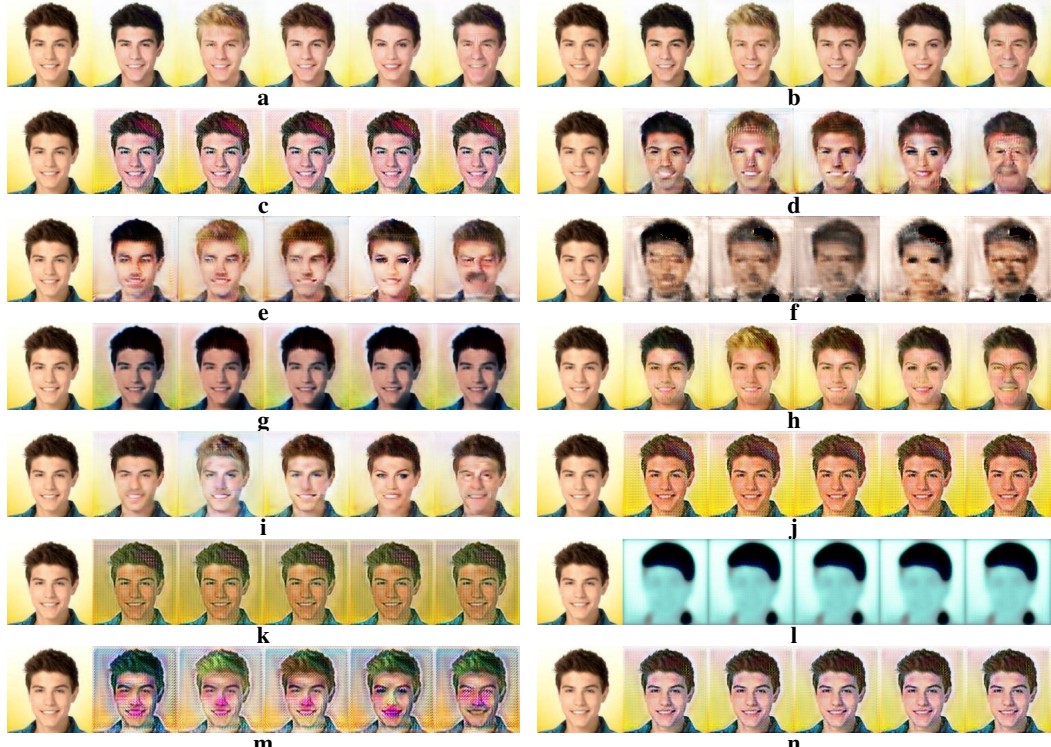

Figure 1: Various approaches to compress StarGAN with network pruning. Each group shows one input face translated with different methods of compressing the network: **a**. Uncompressed, **b**. Self-Supervised **(ours)**, **c**. Small and dense, **d**. One-shot pruning and fine-tuning, **e**. AGP as fine-tuning, **f**. AGP from scratch, **g**. One-shot pruning and distilling, **h**. AGP during distillation, **i**. AGP during distillation with fixed discriminator, **j**. Adversarial learning, **k**. Knowledge distillation, **l**. Distillation on output of intermediate layers, **m**. E-M pruning, and **n**. Prune both G and D models.

good - the compressed generator and discriminator losses converge at 0, just as they did for baseline training. It is clear from the results of querying the generative models (Figures 1h and 1m), though, that this promising convergence is a false positive. In contrast, the curves for our technique predict good performance, and, as we prune more aggressively in Section 6, higher loss values correlate well with worsening FID scores. (Loss curves are provided in the ***Appendix***.)

As pruning and distillation are very effective when compressing models for image classification tasks, why do they fail to compress this generative model? We share three potential reasons:

1. Standard pruning techniques need explicit evaluation metrics; softmax easily reflects the probability distribution and classification accuracy. GANs are typically evaluated subjectively, though some imperfect quantitative metrics have been devised.

2. GAN training is relatively unstable (Arjovsky et al., 2017; Liu et al., 2018a) and sensitive to hyperparameters. The generator and discriminator must be well-matched, and pruning can disrupt this fine balance.

3. The energy of the input and output of a GAN is roughly constant, but other tasks, such as classification, produce an output (1-hot label vector) with much less entropy than the input (three-channel color image of thousands of pixels).

Elaborating on this last point, there is more tolerance in the reduced-information space for the compressed classification model to give the proper output. That is, even if the probability distribution inferred by the original and compressed classification models are not exactly the same, the classified labels *can* be the same. On the other hand, tasks like style-transfer and dataset synthesis have no obvious energy reduction. We need to keep entropy as high as possible (Kumar et al., 2019) during the compression process to avoid mode collapse – generating the same output for different inputs or tasks. Attempting to train a new discriminator to make the compressed generator behave more like the original generator (Wang et al., 2018) suffers from this issue – the new discriminator quickly falls into a low-entropy solution and cannot escape. Not only does this preclude its use on generative tasks, but it means that the compressed network for any task must also be trained from scratch during the distillation process, or the discriminator will never be able to learn.

## 4    SELF-SUPERVISED GENERATOR COMPRESSION

We seek to solve each of the problems highlighted above. Let us restate the general formulation of GAN training: the purpose of the generative model is to generate new samples which are very similar to the real samples, but the purpose of the *discriminative* model is to distinguish between real samples and those synthesized by the generator. A fully-trained discriminator is good at spotting differences, but a well-trained generator will cause it to believe that the a generated sample is both real and generated with a probability of 0.5. Our main insight follows:

By using this powerful discriminator that is already well-trained on the target data set, we can allow it to stand in as a quantitative subjective judge (point 1, above) – if the discriminator can't tell the difference between real data samples and those produced by the compressed generator, then the compressed generator is of the same quality as the uncompressed generator. A human no longer needs to inspect the results to judge the quality of the compressed generator. This also addresses our second point: by starting with a trained discriminator, we know it is well-matched to the generator and will not be overpowered. Since it is so capable (there is no need to prune it to), it also helps to avoid mode collapse. As distillation progresses, it can adapt to and induce fine changes in the compressed generator, which is initialized from the uncompressed generator.

Since the original discriminator is used as a proxy for a human's subjective evaluation, we refer to this as "self-supervised" compression. We illustrate the workflow in Figure 2, using a GAN charged with generating a map image from a satellite image in a domain translation task.

In the right part of Figure 2, the real satellite image ($x$) goes through the original generative model ($\boldsymbol{G}_O$) to produce a fake map image ($\hat{y}_o$). The corresponding generative loss value is $l$-$\boldsymbol{G}_O$. Accordingly, in the left part of Figure 2, the real satellite image ($x$) goes through the compressed generative model ($\boldsymbol{G}_C$) to produce a fake map image ($\hat{y}_c$). The corresponding generative loss value is $l$-$\boldsymbol{G}_C$. This is the inference process of the original and compressed generators, expressed as follows:

$$\hat{y}_o = \boldsymbol{G}_O(x) \tag{1}$$
$$\hat{y}_c = \boldsymbol{G}_C(x) \tag{2}$$

The overall generative difference is measured between the two corresponding generative losses[2]. We use a generative consistent loss function ($\boldsymbol{L}_{GC}$) in the bottom of Figure 2 to represent this process.

$$\boldsymbol{L}_{GC}(l\text{-}\boldsymbol{G}_O, l\text{-}\boldsymbol{G}_C) \to 0 \tag{3}$$

---

[2]In different GANs, the generative loss may consist of several sub-items. For example, StarGAN combines adversarial loss, domain classification loss and reconstruction loss into overall generative loss.

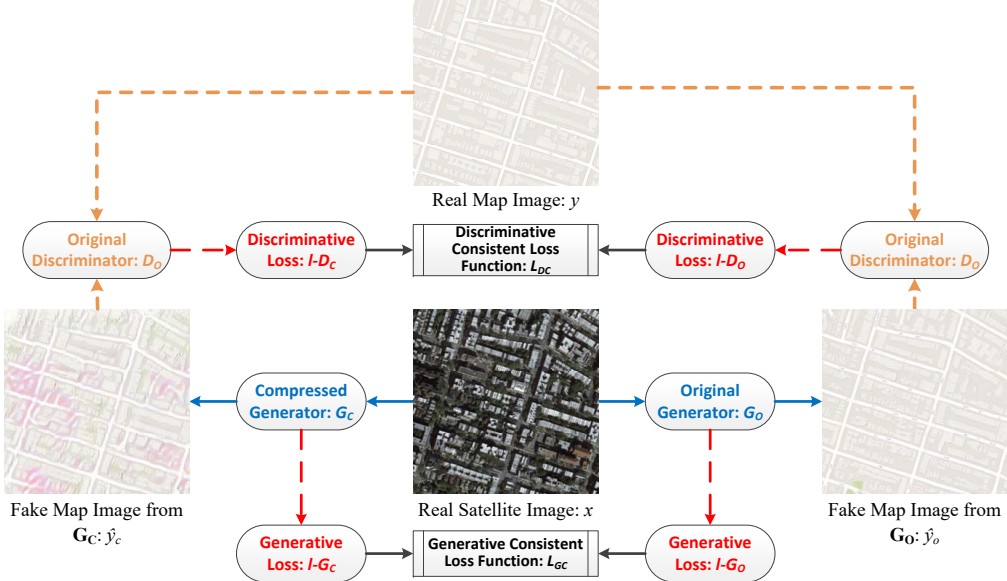

Figure 2: Workflow chart of GAN compression process.

Since the GAN training process aims to reduce the differences between real and generated samples, we stick to this principle in the compression process. In the upper right of Figure 2, real map image ($y$) and fake map image ($\hat{y}_o$) go through the original discriminative model $\boldsymbol{D}_O$. $\boldsymbol{D}_O$ tries to ensure that the distribution of $\hat{y}_o$ is indistinguishable from $y$ using an adversarial loss. The corresponding discriminative loss value is $l\text{-}\boldsymbol{D}_O$. In the upper left of Figure 2, real map image ($y$) and fake map image ($\hat{y}_c$) also go through the original discriminative model $\boldsymbol{D}_O$. In this way, we use the original discriminative model as a "self-supervisor". The corresponding discriminative loss value is $l\text{-}\boldsymbol{D}_C$.

$$l\text{-}\boldsymbol{D}_O = \boldsymbol{D}_O(y, \hat{y}_o) \tag{4}$$

$$l\text{-}\boldsymbol{D}_C = \boldsymbol{D}_O(y, \hat{y}_c) \tag{5}$$

So the discriminative difference is measured between two corresponding discriminative losses. We use the discriminative consistent loss function $\boldsymbol{L}_{DC}$ in the top of Figure 2 to represent this process.

$$\boldsymbol{L}_{DC}(l\text{-}\boldsymbol{D}_O, l\text{-}\boldsymbol{D}_C) \to 0 \tag{6}$$

The generative and discriminative consistent loss functions ($\boldsymbol{L}_{GC}$ and $\boldsymbol{L}_{DC}$) use the weighted normalized Euclidean distance. Taking the **StarGAN** task as the example (other tasks may use different losses):

$$\boldsymbol{L}_{GC}(l\text{-}\boldsymbol{G}_O, l\text{-}\boldsymbol{G}_C) = |l\text{-}\boldsymbol{Gen}_O - l\text{-}\boldsymbol{Gen}_C|/|l\text{-}\boldsymbol{Gen}_O| + \alpha|l\text{-}\boldsymbol{Cla}_O - l\text{-}\boldsymbol{Cla}_C|/|l\text{-}\boldsymbol{Cla}_O| \\ + \beta|l\text{-}\boldsymbol{Rec}_O - l\text{-}\boldsymbol{Rec}_C|/|l\text{-}\boldsymbol{Rec}_O| \tag{7}$$

where $l\text{-}\boldsymbol{Gen}$ is the generation loss term, $l\text{-}\boldsymbol{Cla}$ is the classification loss term, and $l\text{-}\boldsymbol{Rec}$ is the reconstruction loss term. $\alpha$ and $\beta$ are the weight ratios among three loss types. (We use the same values of $\alpha$ and $\beta$ used in the original StarGAN baseline.)

$$\boldsymbol{L}_{DC}(l\text{-}\boldsymbol{D}_O, l\text{-}\boldsymbol{D}_C) = |l\text{-}\boldsymbol{Dis}_O - l\text{-}\boldsymbol{Dis}_C|/|l\text{-}\boldsymbol{Dis}_O| + \delta|l\text{-}\boldsymbol{GP}_O - l\text{-}\boldsymbol{GP}_C|/|l\text{-}\boldsymbol{GP}_O| \tag{8}$$

where $l\text{-}\boldsymbol{Dis}$ is the discriminative loss item, $l\text{-}\boldsymbol{GP}$ is the gradient penalty loss item, and $\delta$ is a weighting factor (again, we use the same value as the baseline).

The overall loss function of GAN compression consists of generative and discriminative differences:

$$L_{Overall} = \boldsymbol{L}_{GC}(l\text{-}\boldsymbol{G}_O, l\text{-}\boldsymbol{G}_C) + \lambda\boldsymbol{L}_{DC}(l\text{-}\boldsymbol{D}_O, l\text{-}\boldsymbol{D}_C), \tag{9}$$

where $\lambda$ is the parameter to adjust the percentages between generative and discriminative losses.

We showed promising results with this method above in the context of prior methods. In the following experiments, we investigate how well the method applies to other networks and tasks (Section 5) and how well the method works on different sparsity ratios and pruning granularities (Section 6).

Table 2: Tasks and networks overview

| Task | Network | Dataset | Resolution | FID Scores when Pruned to | | | | |
|------|---------|---------|------------|-----------|-----|-----|-----|-----|
| | | | | 0% (dense) | 25% | 50% | 75% | 90% |
| Image Synthesis | DCGAN | MNIST | 64x64 | 50.391 | 50.128 | 50.634 | 50.805 | 51.356 |
| Domain Translation | Pix2Pix | Sat → Map | 256x256 | 17.636 | 17.897 | 17.990 | 20.235 | 24.892 |
| Domain Translation | Pix2Pix | Sat ← Map | 256x256 | 30.826 | 30.628 | 30.720 | 34.051 | 38.936 |
| Style Transfer | CycleGAN | Monet → Photo | 256x256 | 63.152 | 63.410 | 63.662 | 66.394 | 70.933 |
| Style Transfer | CycleGAN | Monet ← Photo | 256x256 | 31.987 | 32.102 | 32.346 | 33.913 | 41.409 |
| Image-Image Translation | CycleGAN | Zebra → Horse | 256x256 | 60.930 | 61.005 | 61.102 | 65.898 | 68.450 |
| Image-Image Translation | CycleGAN | Zebra ← Horse | 256x256 | 52.862 | 52.631 | 52.688 | 58.356 | 63.274 |
| Image-Image Translation | StarGAN | CelebA | 128x128 | 6.113 | 6.307 | 6.929 | 6.714 | 7.144 |
| Super Resolution | SRGAN | DIV2K | ≥ 512x512 | 14.653 | 15.236 | 16.609 | 17.548 | 18.376 |

## 5 APPLICATION TO NEW TASKS AND NETWORKS

For the experiments in this section, we choose to prune individual weights in the generator. The final sparsity rate is 50% for all convolution and deconvolution layers in the generator (more aggressive sparsities are discussed in Section 6). Following AGP (Zhu & Gupta, 2018), we gradually increase the sparsity from 5% at the beginning to our target of 50% halfway through the self-supervised training process, and we set the loss adjustment parameter $\lambda$ to 0.5 in all experiments. We use PyTorch (Paszke et al., 2017), implement the pruning and training schedules with Distiller (Zmora et al., 2018), and train and generate results with a V100 GPU (NVIDIA, 2017) using FP32 to match public baselines. In all experiments, the data sets, data preparation, and baseline training all follow from the public repositories - details are summarized in Table 2. We start by assuming an extra 10% of the original number of epochs will be required; in some cases, we reduced the overhead to only 1% while maintaining subjective quality. We include representative results for each task, but a more comprehensive collection of outputs for each experiment is included in the ***Appendix***.

**Image Synthesis**. We apply the proposed compression method to DCGAN (Radford et al., 2016)[3], a network that learns to synthesize novel images belonging to a given distribution. We task DCGAN with generating images that could belong to the MNIST data set, with results shown in Figure 3.

**Domain Translation**. We apply the proposed compression method to pix2pix (Isola et al., 2017)[4], an approach to learn the mapping between paired training examples by applying conditional adversarial networks. In our experiment, the task is synthesizing fake satellite images from label maps and vice-versa. Representative results of this bidirectional task are shown in Figure 4.

**Style Transfer**. We apply the proposed compression method to CycleGAN (Zhu et al., 2017a), used to exchange the style of images from a source domain to a target domain in the absence of paired training examples. In our experiment, the task is to transfer the style of real photos with that

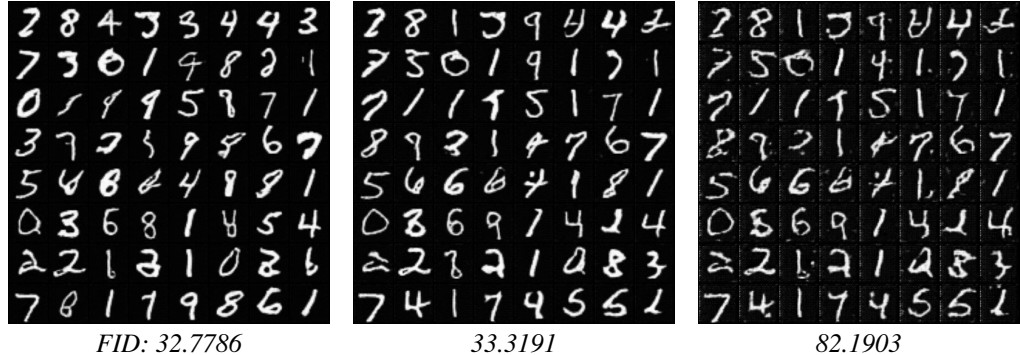

*FID: 32.7786*          *33.3191*          *82.1903*

Figure 3: Image synthesis on MNIST dataset with DCGAN. Columns 1-3: Handwritten numbers generated by the original generator, pruned generator of 50%, 75% fine-grained sparsity.

---

[3]DCGAN baseline: `https://github.com/pytorch/examples/tree/master/dcgan`.
[4]Pix2pix, CycleGAN: `https://github.com/junyanz/pytorch-CycleGAN-and-pix2pix`.

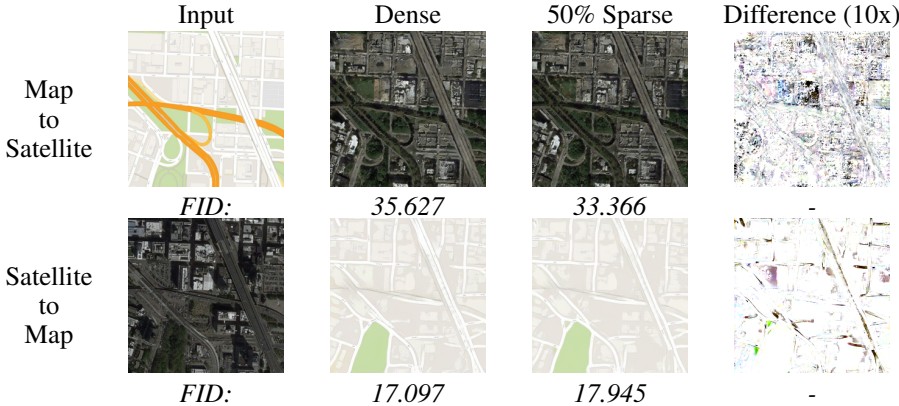

Figure 4: Representative results for domain translation: pix2pix.

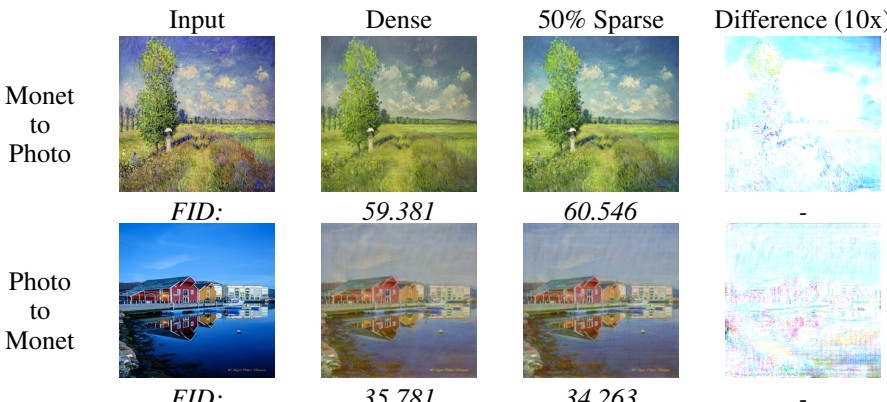

Figure 5: Representative results for style transfer: CycleGAN.

of the Monet's paintings. Representative results of this bidirectional task are shown in Figure 5: photographs are given the style of Monet's paintings and vice-versa.

**Image-to-image Translation**. In addition to the StarGAN results above (Section 3, Figure 1), we apply the proposed compression method to CycleGAN (Zhu et al., 2017a) performing bidirectional translation between zebra and horse images. Results are shown in Figure 6.

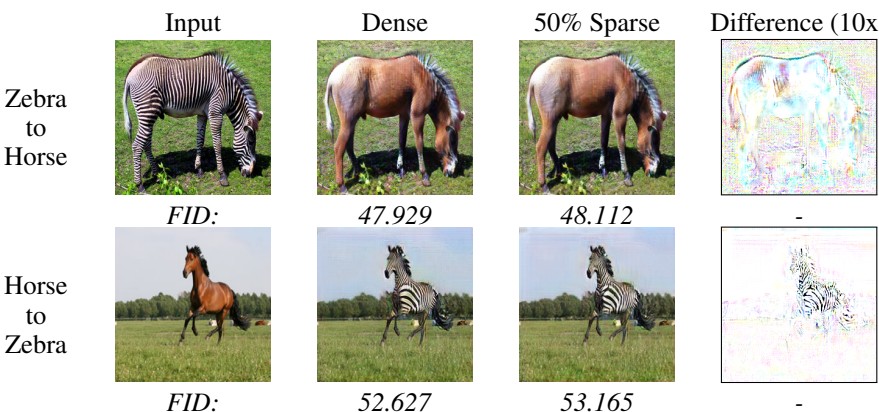

Figure 6: Representative image-to-image translation results: CycleGAN.

Table 3: *PSNR* (dB), *SSIM* and *FID* indicators for Validation Datasets

| Dataset | Original Generator | | | Filter-Compressed G | | | Element-Compressed G | | |
|---|---|---|---|---|---|---|---|---|---|
| | PSNR | SSIM | FID | PSNR | SSIM | FID | PSNR | SSIM | FID |
| Set5 | 30.063393 | 0.852733 | 30.761999 | 30.234316 | 0.859817 | 35.514204 | 30.484014 | 0.862475 | 36.824148 |
| Set14 | 26.643850 | 0.716294 | 55.457409 | 27.314664 | 0.744525 | 82.118059 | 27.417112 | 0.744101 | 70.125821 |
| DIV2K_Validation | 28.205665 | 0.778364 | 14.653151 | 28.875953 | 0.800625 | 18.499896 | 28.974868 | 0.800767 | 16.608606 |

Figure 7: Representative super resolution results: SRGAN (with enlargements of boxed areas).

**Super Resolution**. We apply self-supervised compression to SRGAN (Ledig et al., 2017)[5], which uses a discriminator network trained to differentiate between upscaled and the original high-resolution images. We trained SRGAN on the DIV2K data set Agustsson & Timofte (2017), and use the DIV2K validation images, as well as Set5 Bevilacqua et al. (2012) and Set14 Zeyde et al. (2010) to report deployment quality. In this task, quality is often evaluated by two metrics: Peak Signal-to-Noise Ratio (PSNR) (Huynh-Thu & Ghanbari, 2008) and Structural Similarity (SSIM) (Wang et al., 2004). We also show FID scores (Heusel et al., 2017) for our results in the results summarized in Table 3, and a representative output is shown in Figure 7. These results also include filter-pruned generators (see Section 6).

## 6 EFFECT OF PRUNING RATIO AND GRANULARITY

After showing that self-supervised compression applies to many tasks and networks with a moderate, fine-grained sparsity of 50%, we expand the scope of the investigation to include different pruning granularities and rates. From coarse to fine, we can compress and remove the entire filters (3D-level), kernels (2D-level), vectors (1D-level) or individual elements (0D-level). In general, finer-grained pruning results in higher accuracy for a given sparsity rate, but coarser granularities are easier to exploit for performance gains due to their regular structure. Similarly, different sparsity rates, leaving many nonzero weights or few, can result in varying levels of quality in the final network.

We pruned all tasks by removing both single elements (0D) and entire filters (3D). Further, for each granularity, we pruned to final sparsities of 25%, 50%, 75%, and 90%. Representative results for CycleGAN (Monet → Photo) are shown in Figure 8, but in general, 0D pruning is less invasive, even at higher sparsities. Up to 90% fine-grained sparsity, some fine details faded away in pix2pix, but filter pruning results in drastic color shifts and loss of details at even 25% sparsity.

## 7 CONCLUSION AND FUTURE WORK

In this paper, we propose using a pre-trained discriminator to self-supervise the compression of a generative adversarial network. We show that it is effective and applies to many tasks commonly solved with GANs, unlike traditional compression approaches. Comparing the compressed generators with the baseline models on different tasks, we can conclude that the compression method

---

[5]SRGAN baseline implementation: `https://github.com/xinntao/BasicSR`.

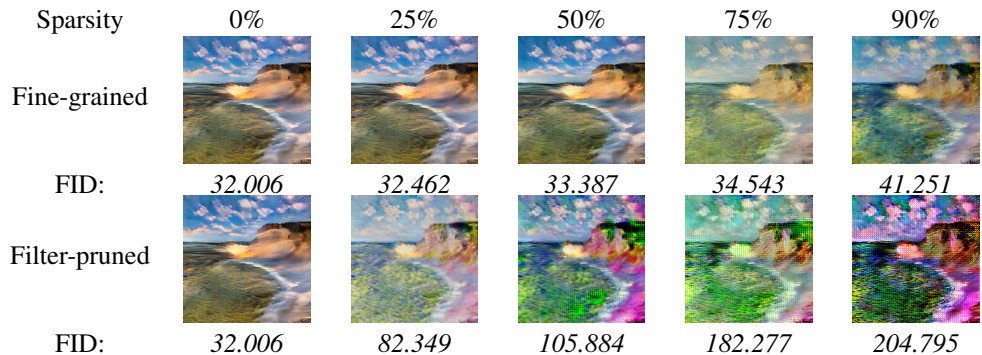

| Sparsity | 0% | 25% | 50% | 75% | 90% |
|---|---|---|---|---|---|

Fine-grained

| FID: | *32.006* | *32.462* | *33.387* | *34.543* | *41.251* |
|---|---|---|---|---|---|

Filter-pruned

| FID: | *32.006* | *82.349* | *105.884* | *182.277* | *204.795* |
|---|---|---|---|---|---|

Figure 8: Representative results for pruning rate and granularity study of style transfer.

performs well both in subjective and quantitative evaluations. Advantages of the proposed method include:

- The results from the compressed generators are greatly improved over past work.
- The self-supervised compression is much shorter than the original GAN training process. It only takes 1%-10% training effort to get an optimal compressed generative model.
- It is an end-to-end compression schedule that does not require objective evaluation metrics.
- We introduce a single optional hyperparameter (fixed to 0.5 for all our experiments).

We use self-supervised GAN compression to show that pruning whole filters, which can work well for image classification models (Li et al., 2017), may perform poorly for GAN applications. Even pruned at a moderate sparsity (e.g. 25% in Figure 8), the generated image has an obvious color shift and does not transfer the photorealistic style. In contrast, the fine-grained compression stategy works well for all tasks we explored. SRGAN seems to be an exception to filter-pruning's poor results; we have to look closely to see differences, and it's not clear which is subjectively better.

We have not tried to achieve extremely aggressive compression rates with complicated pruning strategies. Different models may be able to tolerate different amounts of pruning when applied to a task, which we leave to future work. Similarly, we have used network pruning to show the importance and utility of the proposed method, but self-supervised compression is general to other techniques, such as quantization, weight sharing, etc. There are other tasks for which GANs can provide compelling results, and newer networks for tasks we have already explored; future work will extend our self-supervised compression method to these new areas. Finally, self-supervised compression may apply to other network types and tasks if a discriminator is trained alongside the teacher and student networks.

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

## A  APPENDIX

Since GANs are typically judged with subjective, qualitative observations, we present several results for each of the experiments in the main paper so readers can see the motivation for the conclusions drawn therein. We organize this document in the following way:

- Section A.1: Naïve Compression with StarGAN
- Section A.2: Image Synthesis with DCGAN
- Section A.3: Domain Translation with Pix2Pix
- Section A.4: Style Transfer with CycleGAN
- Section A.5: Image-Image Translation with CycleGAN
- Section A.6: Image-Image Translation with StarGAN
- Section A.7: Super Resolution with SRGAN
- Section A.8: Effect of Sparsity Granularity and Ratio

## A.1 NAÏVE COMPRESSION: STARGAN

The loss curves for the comparative experiment in Figure 1 are shown in Figure 9.

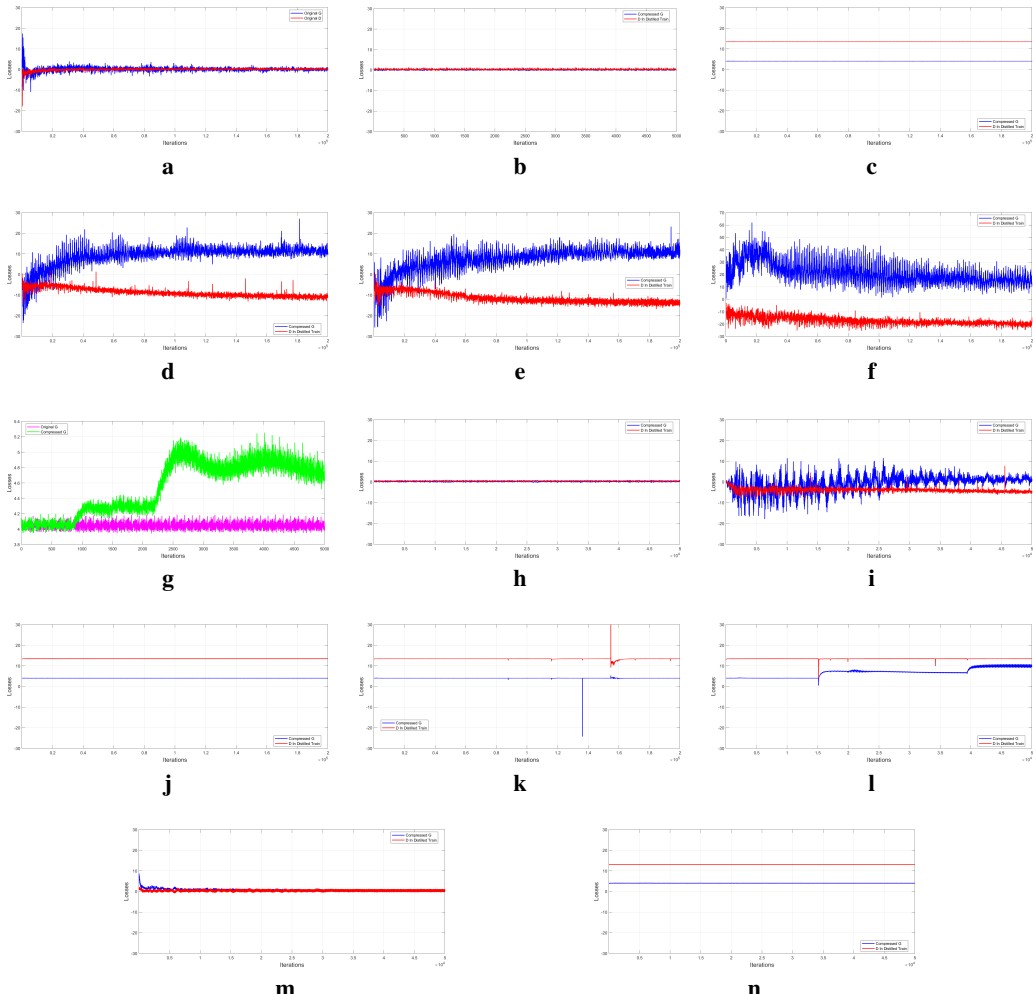

Figure 9: Loss curves of image-to-image translation pruning. **(a)**. Loss curve of StarGAN baseline. Loss curve of training the compressed generator from **(b)**. Self-Supervised fine-tuning **(ours)**, **(c)**. Smaller dense network, **(d)**. One-shot pruning and naive fine-tuning, **(e)**. Gradual pruning and naive fine-tuning, **(f)**. Gradual pruning during the initial training, **(g)**. One-shot pruning and distillation as fine-tuning, **(h)**. Gradual pruning and distillation as fine-tuning, **(i)**. AGP as fine-tuning and distillation without fine-tuning the original loss, **(j)**. Adversarial learning (fine-tuning), **(k)**. Knowledge distillation, **(l)**. Distillation on output of intermediate layers, **(m)**. E-M Quantization, and **(n)**. Prune both G and D models. **(a)**, **(c)** and **(f)** start from a randomly-initialized network at epoch 0, others pick up at the end of **(a)**.

Figures 10-12 show outputs of StarGAN compressed with various existing techniques (**c-n**), and the proposed self-supervised method (**b**). The baseline output is at the top (**a**) of each figure for comparison. Each row shows one input face translated to have black hair, blond hair, brown hair, the opposite gender, and a different age, and each row is a different method of compressing the network (the key is identical to that of Figure 9).

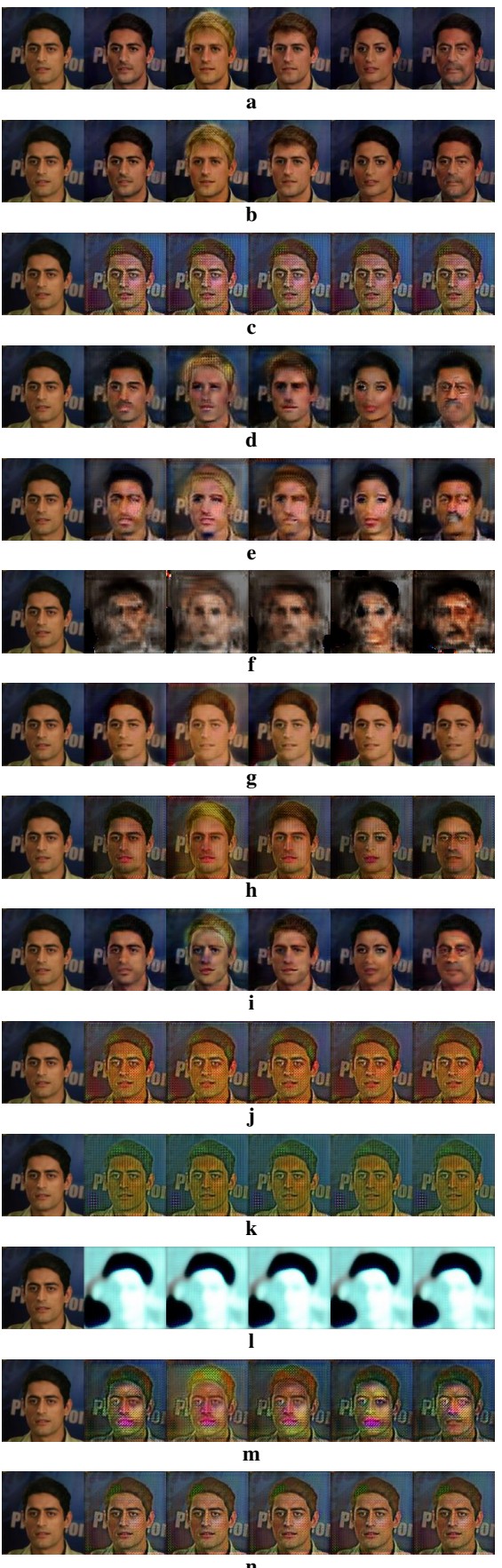

Figure 10: Example 1 of various approaches to compress StarGAN.

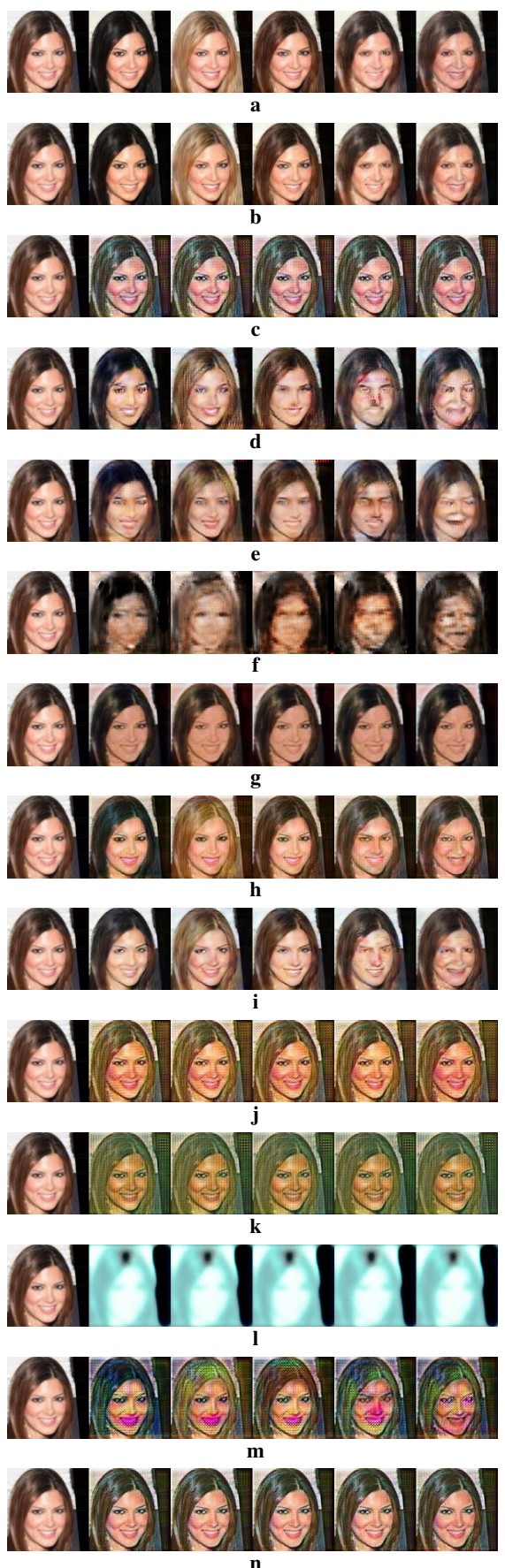

Figure 11: Example 2 of various approaches to compress StarGAN.

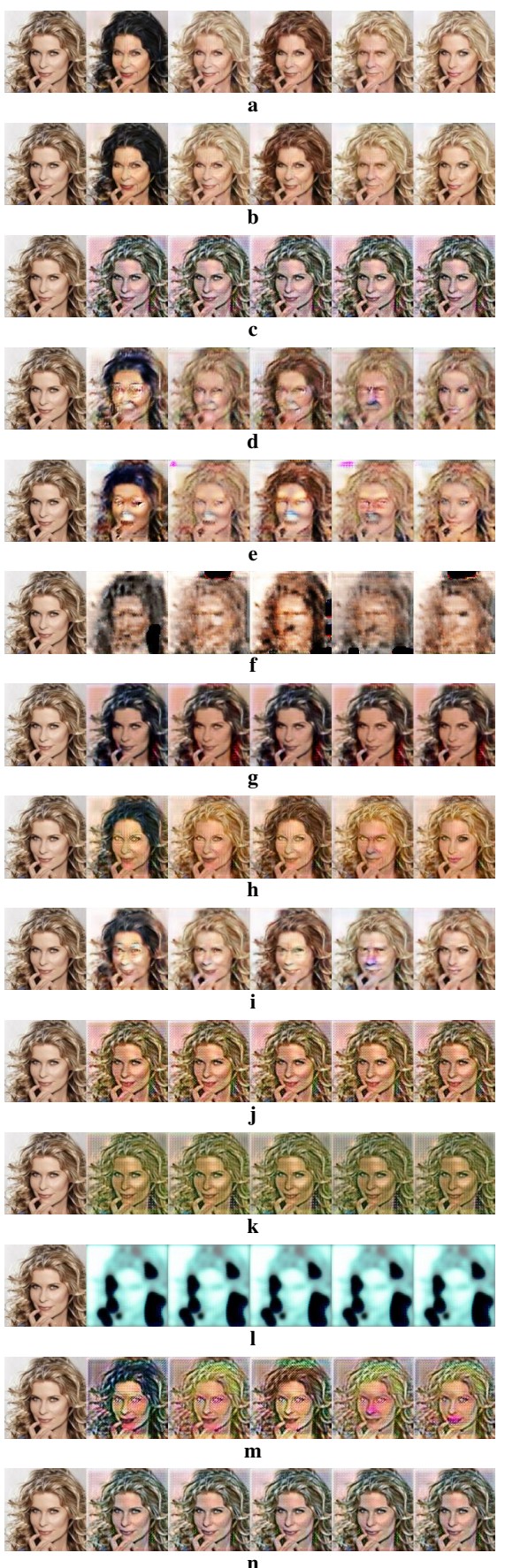

Figure 12: Example 3 of various approaches to compress StarGAN.

A.2 IMAGE SYNTHESIS: DCGAN (50% AND 75% FINE-GRAINED SPARSITY)

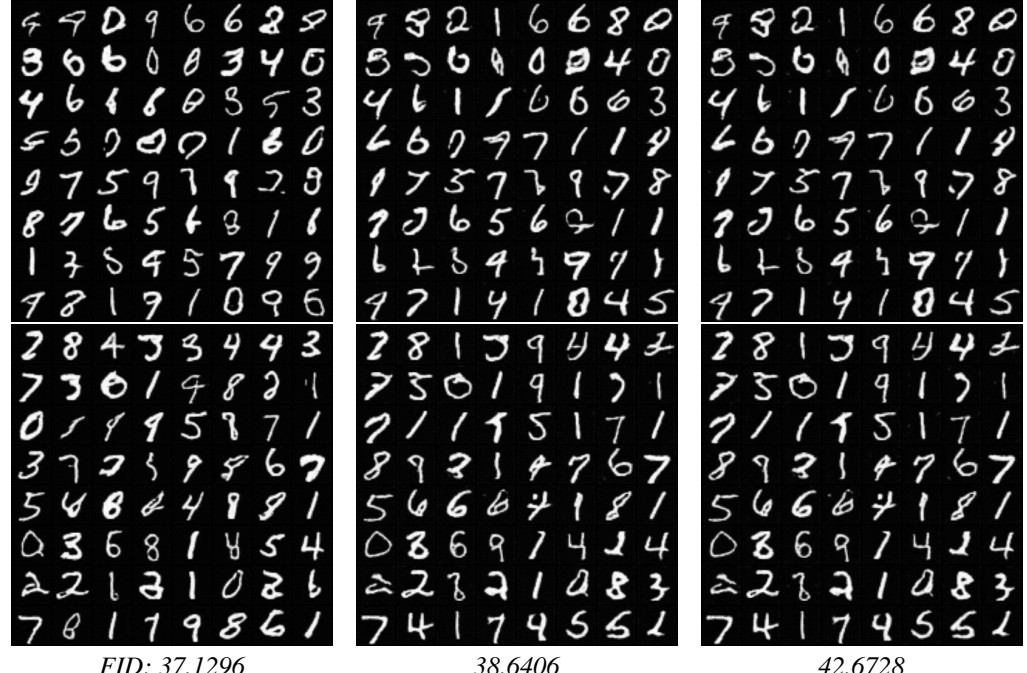

*FID: 37.1296*          *38.6406*          *42.6728*

Figure 13: Image synthesis on MNIST dataset with DCGAN pruned to 50% with fine-grained sparsity. Column 1: Handwritten numbers generated by the original generator, 2: Handwritten numbers generated by the generator pruned with our method, 3: Handwritten numbers generated by the pruned generator with traditional knowledge distillation adapted for GANs (Aguinaldo et al., 2019).

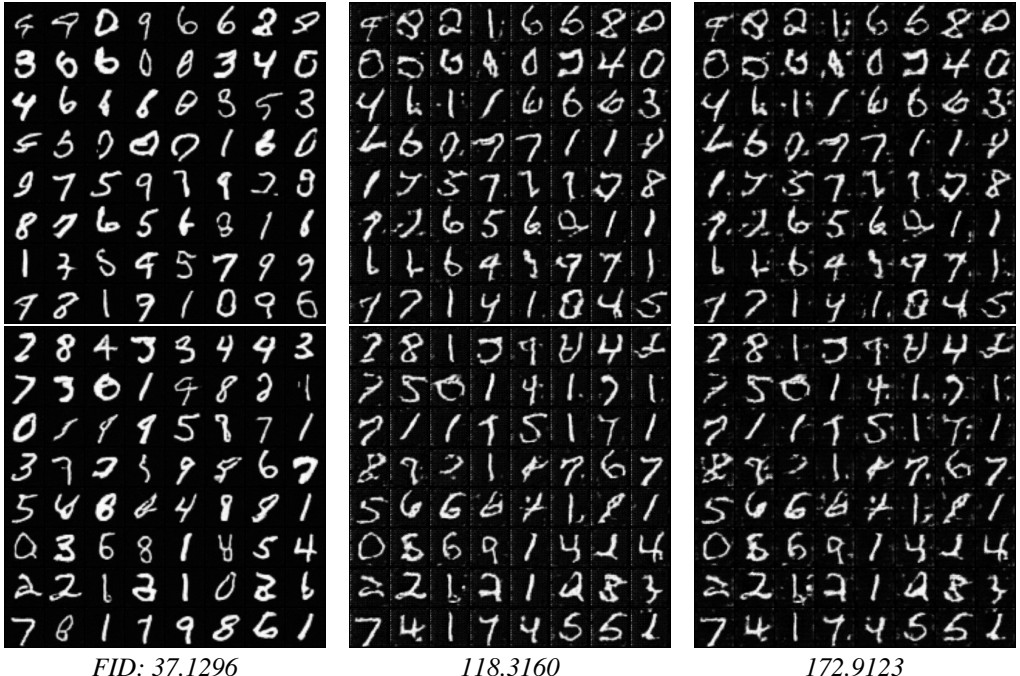

*FID: 37.1296*          *118.3160*          *172.9123*

Figure 14: Image synthesis on MNIST dataset with DCGAN of 75% fine-grained sparsity. Column 1: Handwritten numbers generated by the original generator, 2: Handwritten numbers generated by the generator pruned with our method, Column 3: Handwritten numbers generated by the pruned generator with traditional knowledge distillation adapted for GANs.

### A.3 DOMAIN TRANSLATION: PIX2PIX (50% FINE-GRAINED SPARSITY)

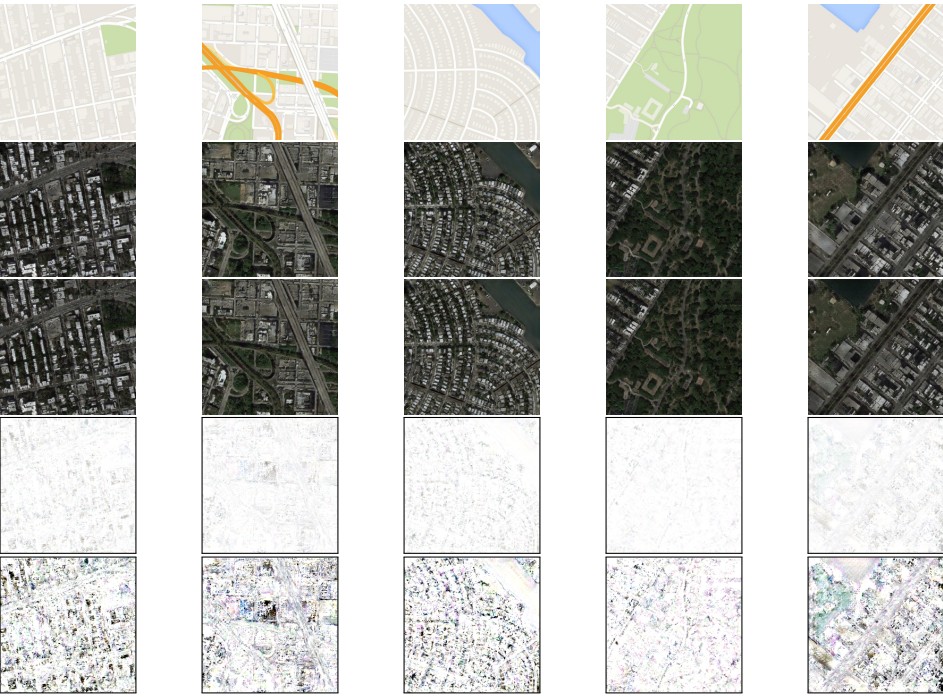

Figure 15: Image synthesis: from label maps to fake satellite images. Row 1: Original label maps, Row 2: Satellite images generated by the original generator, Row 3: Satellite images generated by the pruned generator, Row 4: Residual difference between generated images in Row 2 and 3, Row 5: Residuals amplified by 10x.

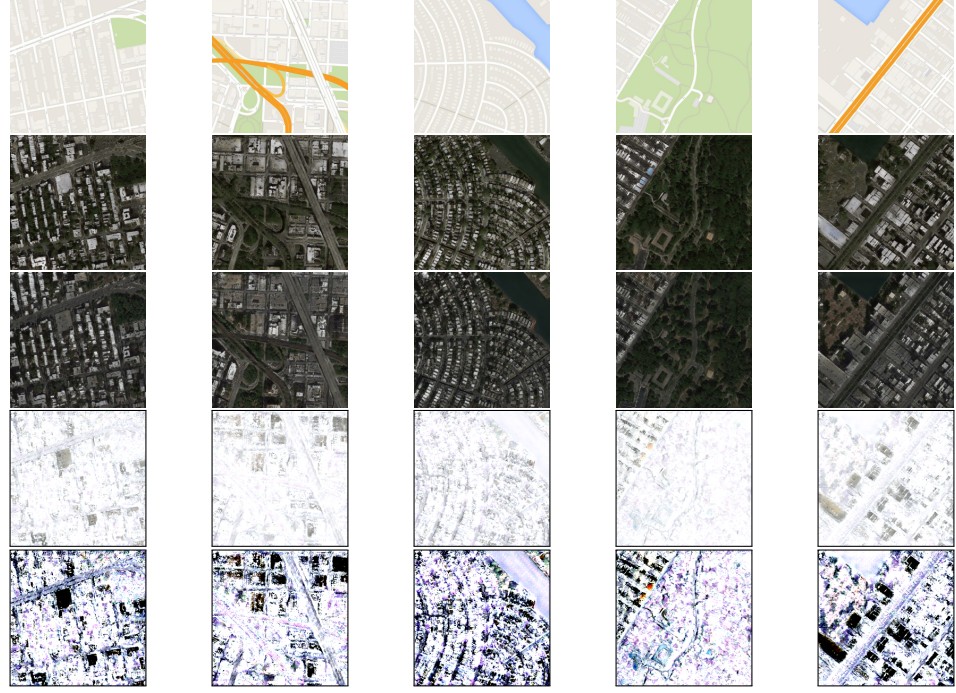

Figure 16: Image synthesis: Two different random seeds, **unpruned**. Row 1: Original label maps, Rows 2-3: Generated fake satellite images by original generator trained with random seeds 15 and 63, Row 4: Residual difference between generated images in Row 2 and 3. Row 5: Residuals amplified by 10x for higher contrast.

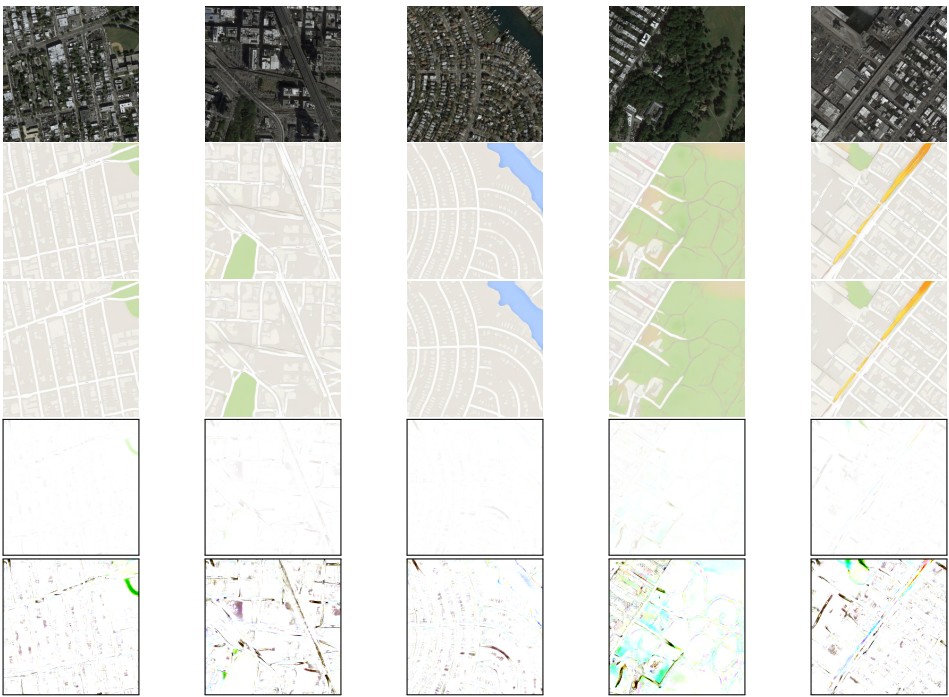

Figure 17: Image synthesis: from satellite images to fake label maps. Row 1: Original satellite images, Row 2: Label maps generated by the original generator, Row 3: Label maps generated by the pruned generator, Row 4: Residual difference between generated images in Row 2 and 3, Row 5: Residuals amplified by 10x.

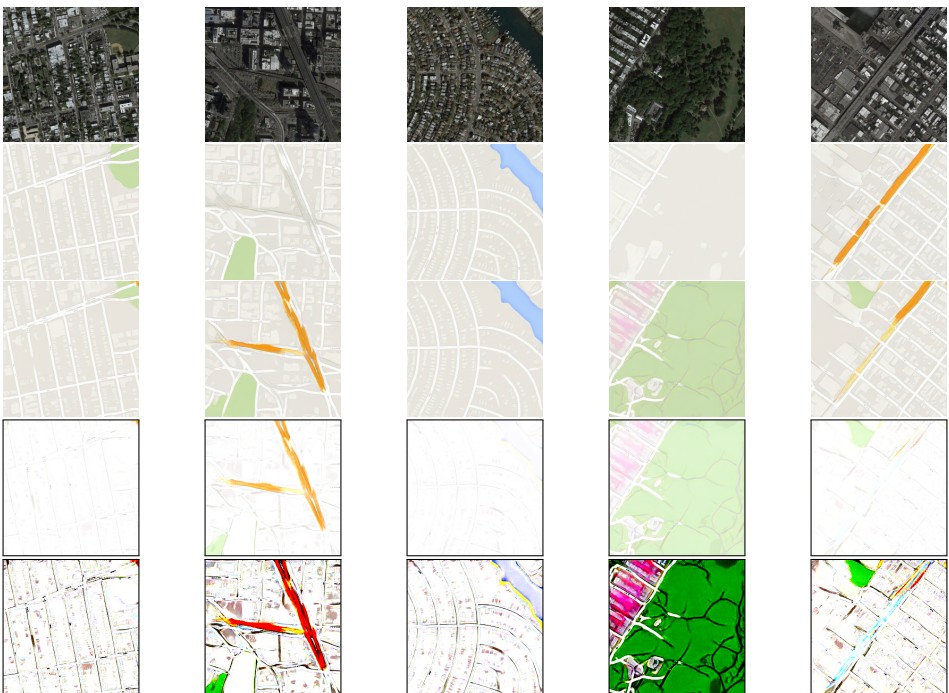

Figure 18: Image synthesis: Two different random seeds, **unpruned**. Row 1: Original satellite images, Rows 2-3: Generated fake label maps by original generator trained with random seeds 15 and 63, Row 4: Residual difference between generated images in Row 2 and 3. Row 5: Residuals amplified by 10x for higher contrast.

## A.4   STYLE TRANSFER: CYCLEGAN (50% FINE-GRAINED SPARSITY)

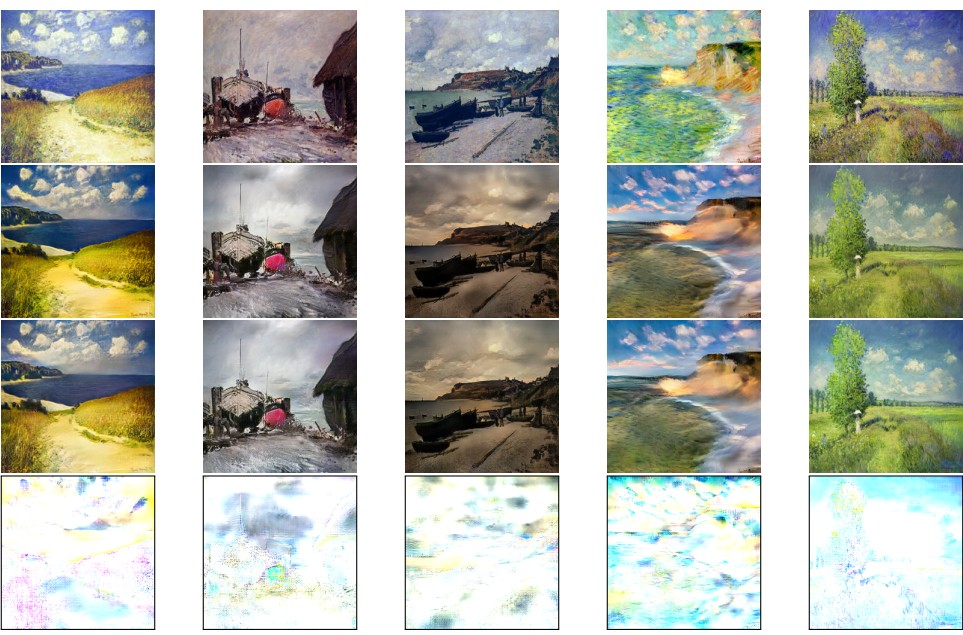

Figure 19: Style transfer: from Monet to real photo style. Row 1: Original artwork images from Monet, Row 2: photographic style applied by the original generator, Row 3: photographic style applied by the compressed generator, Row 4: Residual difference between style transferred images in Row 2 and 3, amplified by 10x.

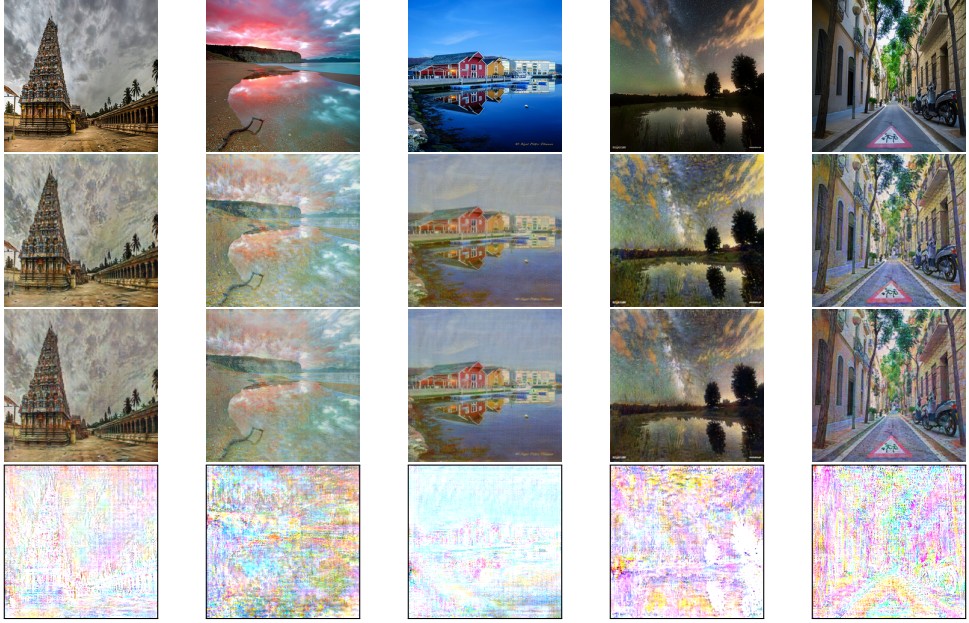

Figure 20: Style transfer: from real photo to Monet artwork style. Row 1: Original photos, Row 2: Monet's style applied by the original generator, Row 3: Monet's style applied by the compressed generator, Row 4: Residual difference between style transferred images in Row 2 and 3 amplified by 10x.

A.5    IMAGE-IMAGE TRANSLATION: CYCLEGAN (50% FINE-GRAINED SPARSITY)

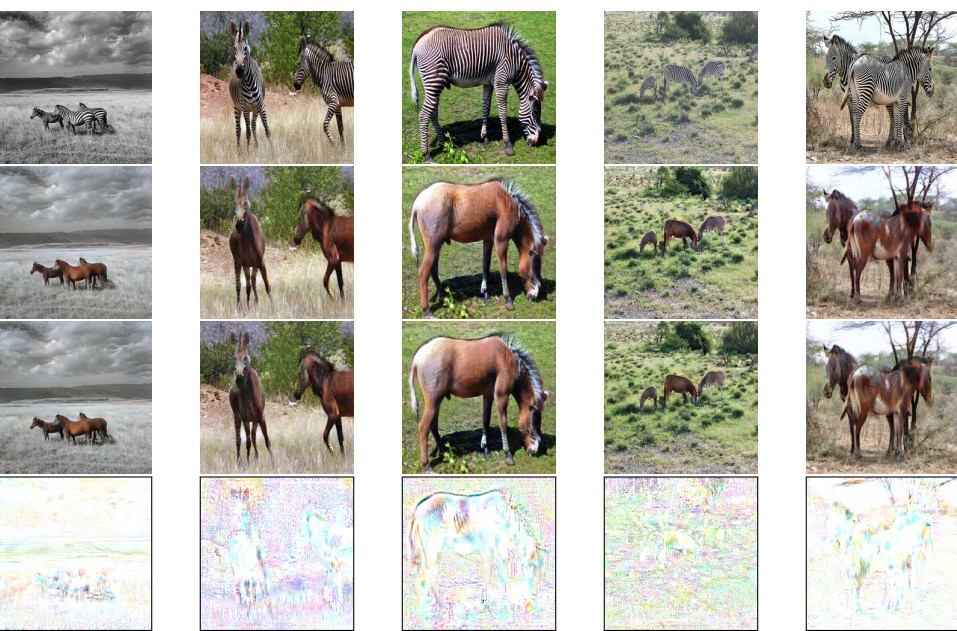

Figure 21: Image-to-image translation experiment: from real zebra images to fake horse images. Row 1: Original real zebra images, Row 2: Corresponding translated horse images by original generator, Row 3: Translated horse images by compressed generator, Row 4: Residual difference between translated images in Row 2 and 3 amplified by 10x.

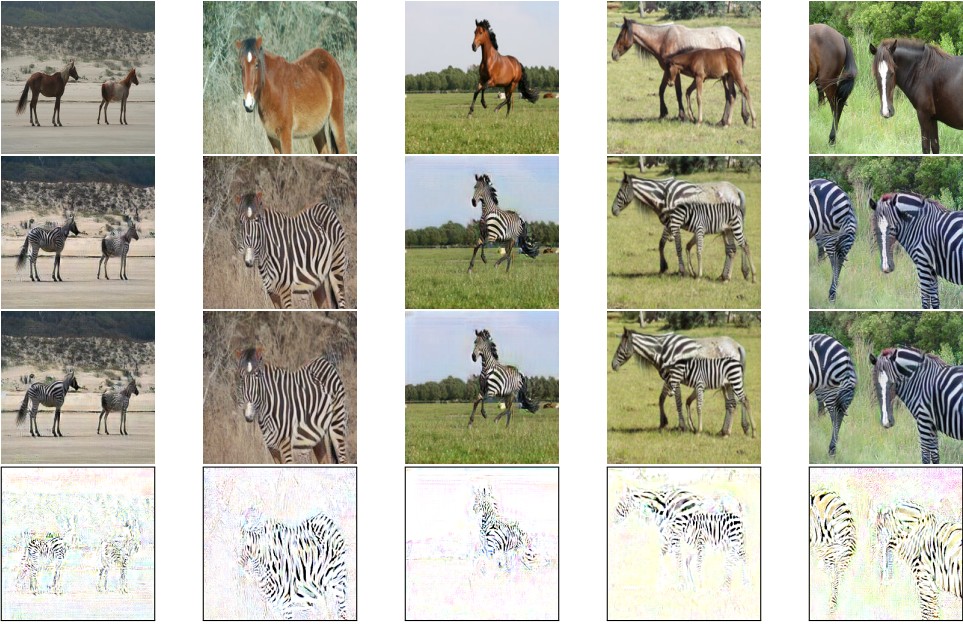

Figure 22: Image-to-image translation experiment: from real horse images to fake zebra images. Row 1: Original real horse images, Row 2: Corresponding translated zebra images by original generator, Row 3: Translated zebra images by compressed generator, Row 4: Residual difference between translated images in Row 2 and 3 amplified by 10x.

A.6    IMAGE-IMAGE TRANSLATION: STARGAN (50% FINE-GRAINED SPARSITY)

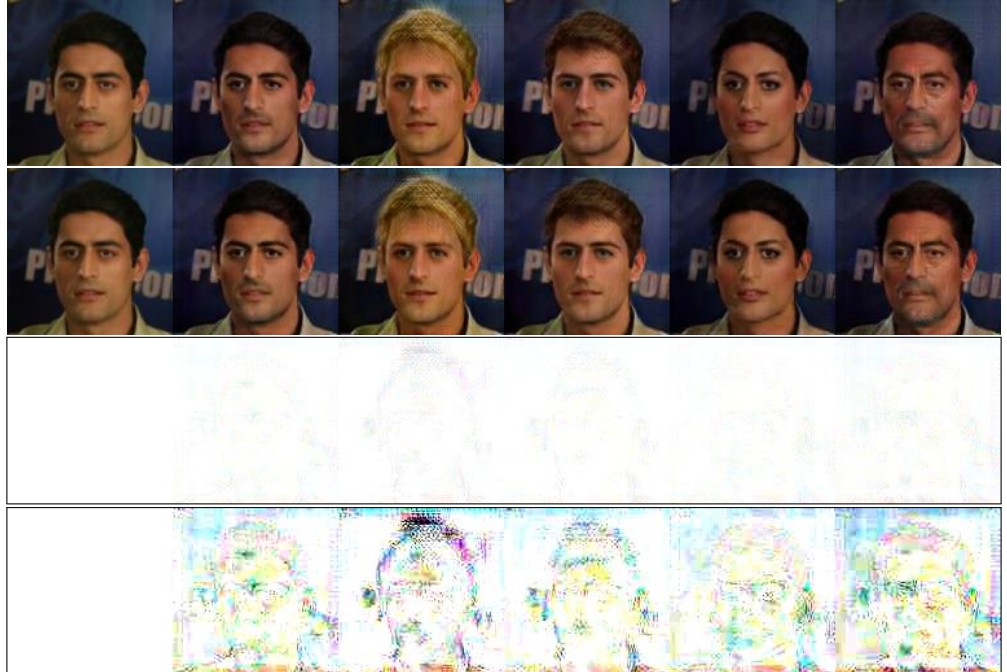

Figure 23: Image-to-image translation example 1: facial attribute translation. Columns: 1. Original facial images, 2-4. Translated images to (black, blond, brown) hair colors, 5. Translated images to other gender, 6. Translated images to other age. Rows: Images translated by 1. original generator and 2. compressed generator, 3. Residual difference between Rows 1 and 2, 4. Residuals amplified by 10x.

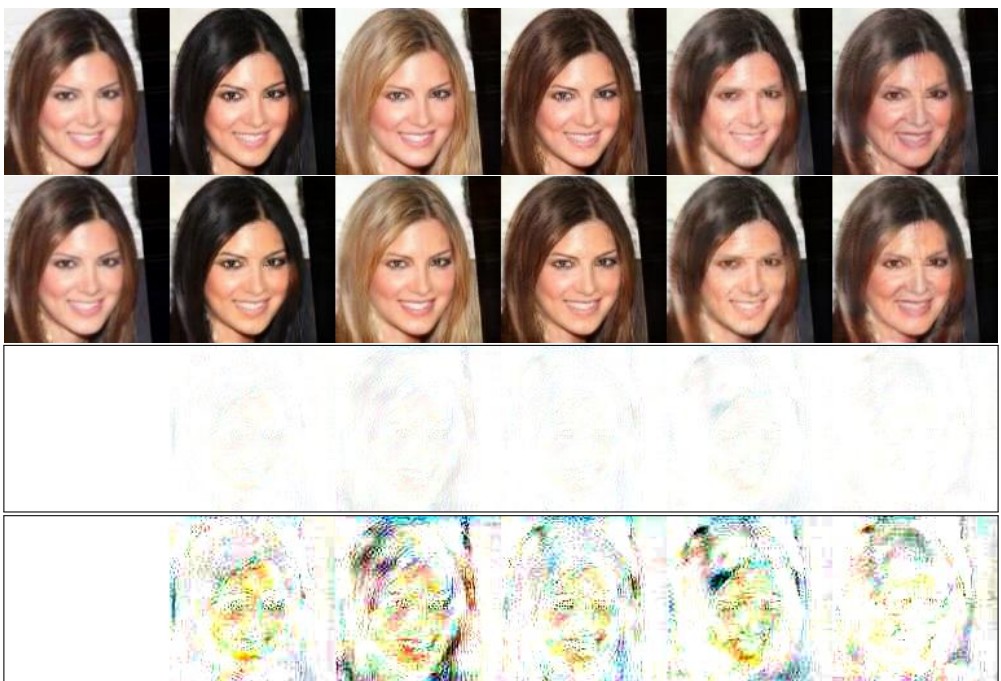

Figure 24: Image-to-image translation example 2: facial attribute translation. Columns: 1. Original facial images, 2-4. Translated images to (black, blond, brown) hair colors, 5. Translated images to other gender, 6. Translated images to other age. Rows: Images translated by 1. original generator and 2. compressed generator, 3. Residual difference between Rows 1 and 2, 4. Residuals amplified by 10x.

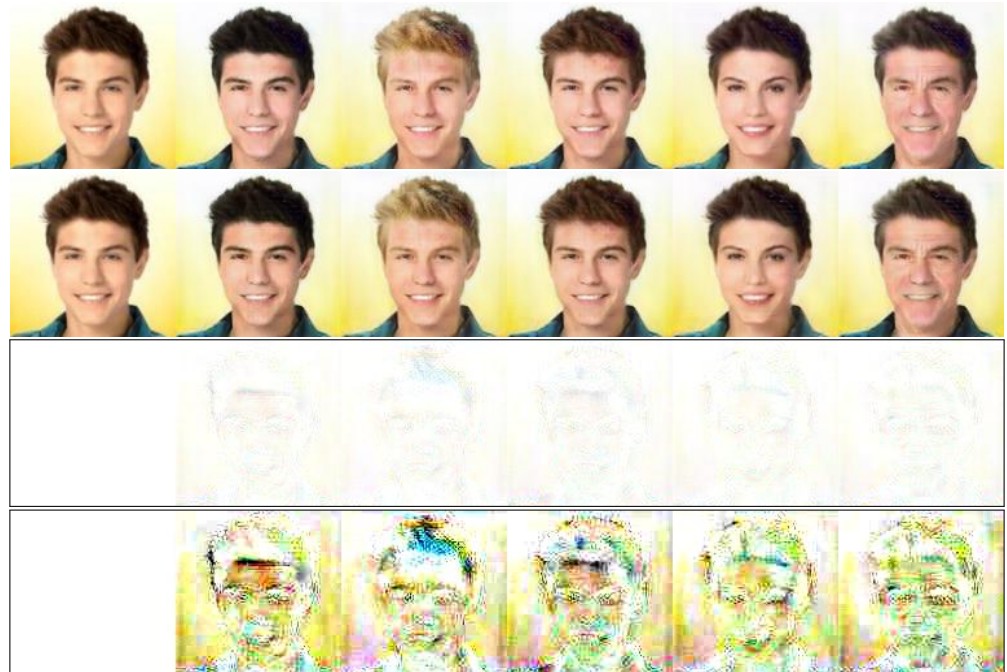

Figure 25: Image-to-image translation example 3: facial attribute translation. Columns: 1. Original facial images, 2-4. Translated images to (black, blond, brown) hair colors, 5. Translated images to other gender, 6. Translated images to other age. Rows: Images translated by 1. original generator and 2. compressed generator, 3. Residual difference between Rows 1 and 2, 4. Residuals amplified by 10x.

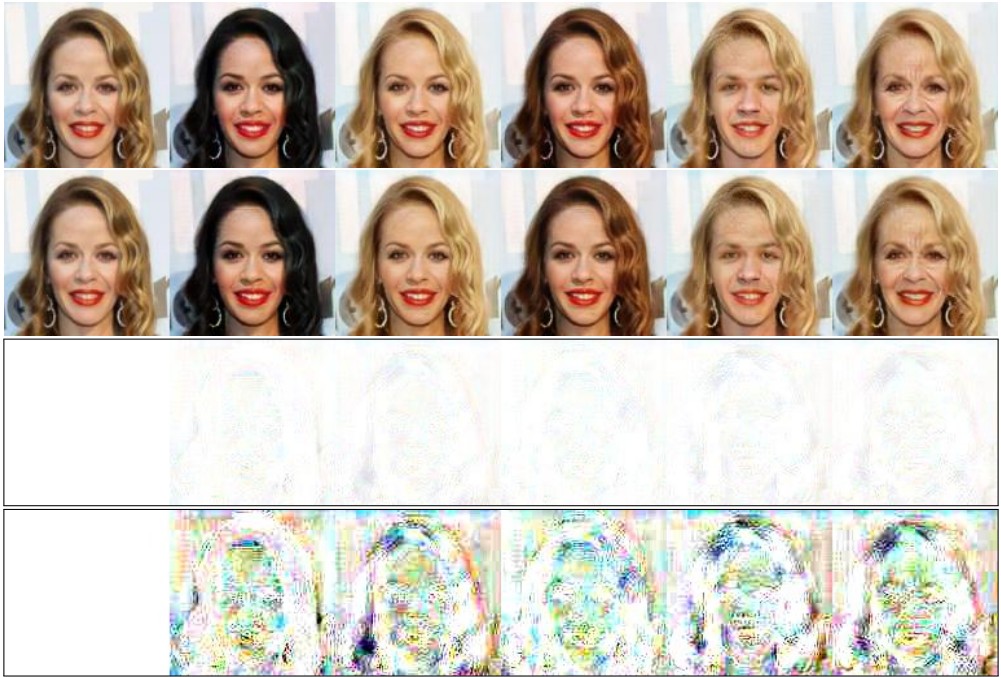

Figure 26: Image-to-image translation example 4: facial attribute translation. Columns: 1. Original facial images, 2-4. Translated images to (black, blond, brown) hair colors, 5. Translated images to other gender, 6. Translated images to other age. Rows: Images translated by 1. original generator and 2. compressed generator, 3. Residual difference between Rows 1 and 2, 4. Residuals amplified by 10x.

A.7   SUPER RESOLUTION: SRGAN (50% FINE-GRAINED AND FILTER-PRUNED SPARSITY)

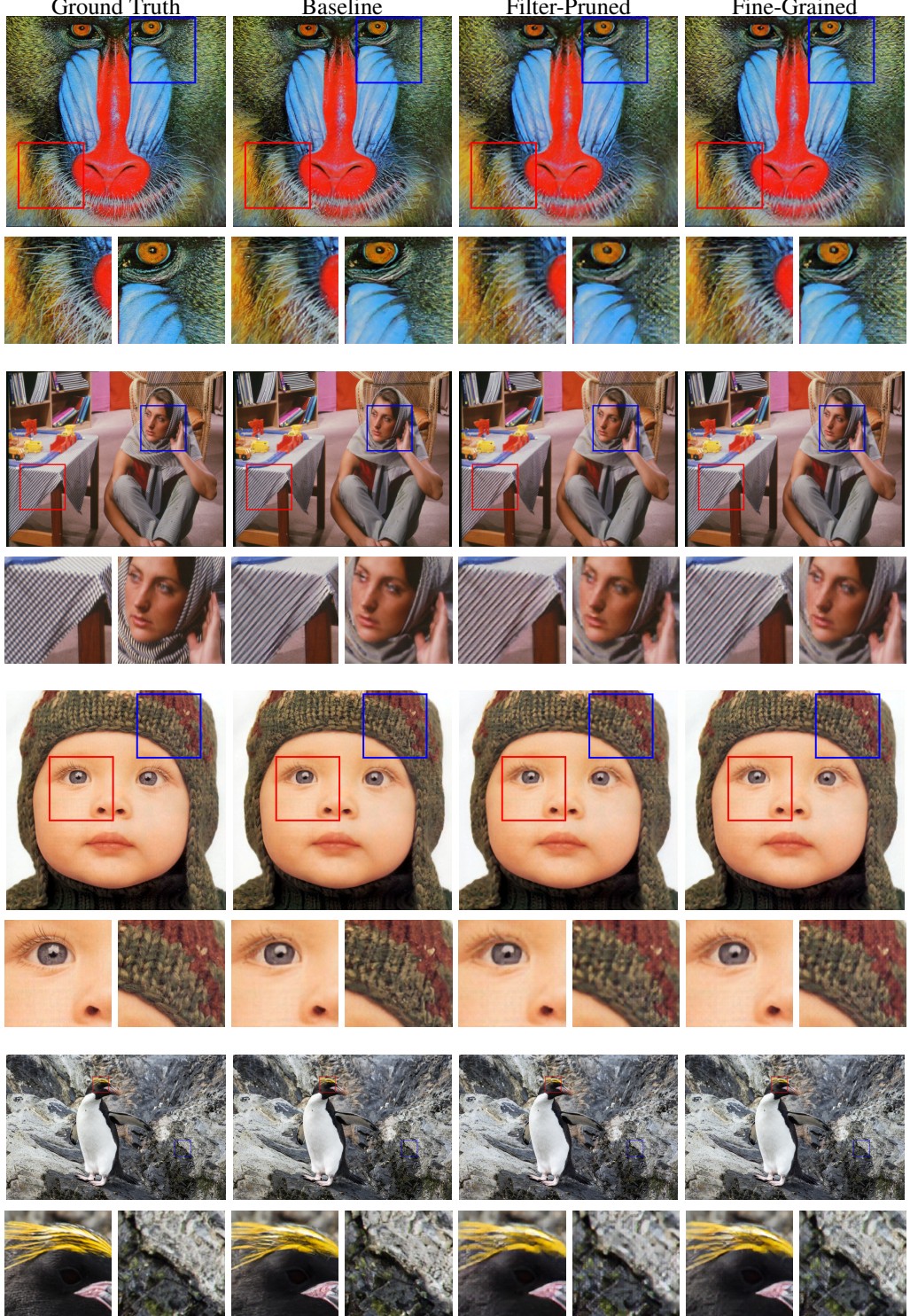

Figure 27: Super resolution experiment. Column 1: Original high resolution images, Columns 2-4: Corresponding generated real high resolution images by original, filter-compressed, element-compressed generators. Each second row provides a detailed view of boxed regions.

## A.8 EFFECT OF SPARSITY GRANULARITY AND RATIO

**Pix2Pix**: map to satellite domain translation.

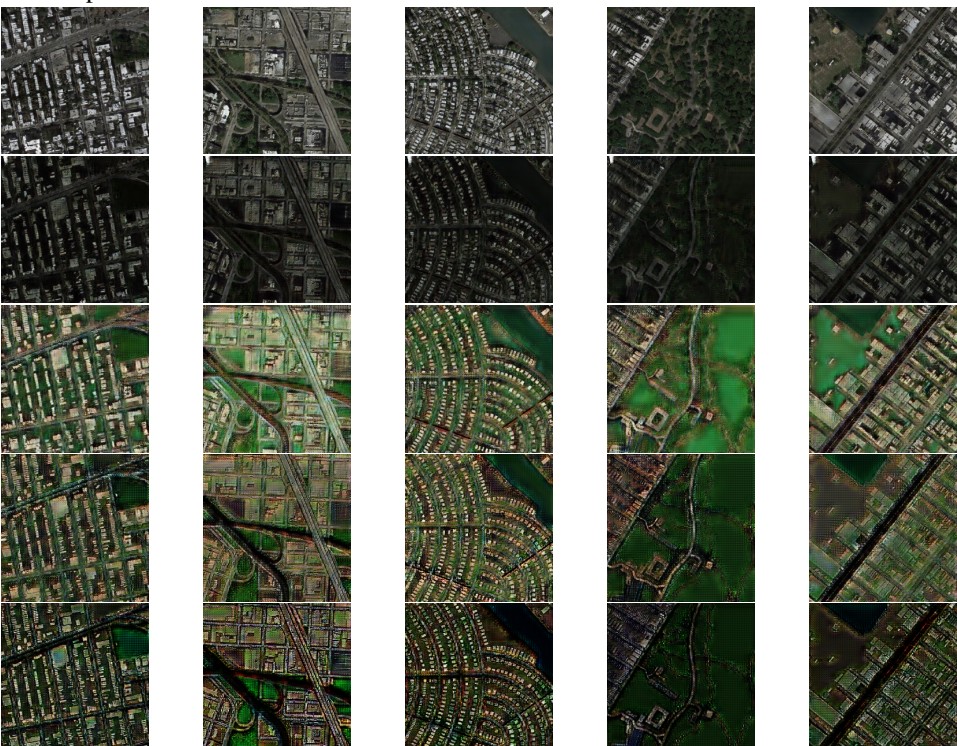

Figure 28: Domain translation: filter pruning to different sparsity levels. Row 1: Output of the baseline generator. Rows 2-5: Synthesized satellite images by generators pruned to sparsities of 25%, 50%, 75%, 90%.

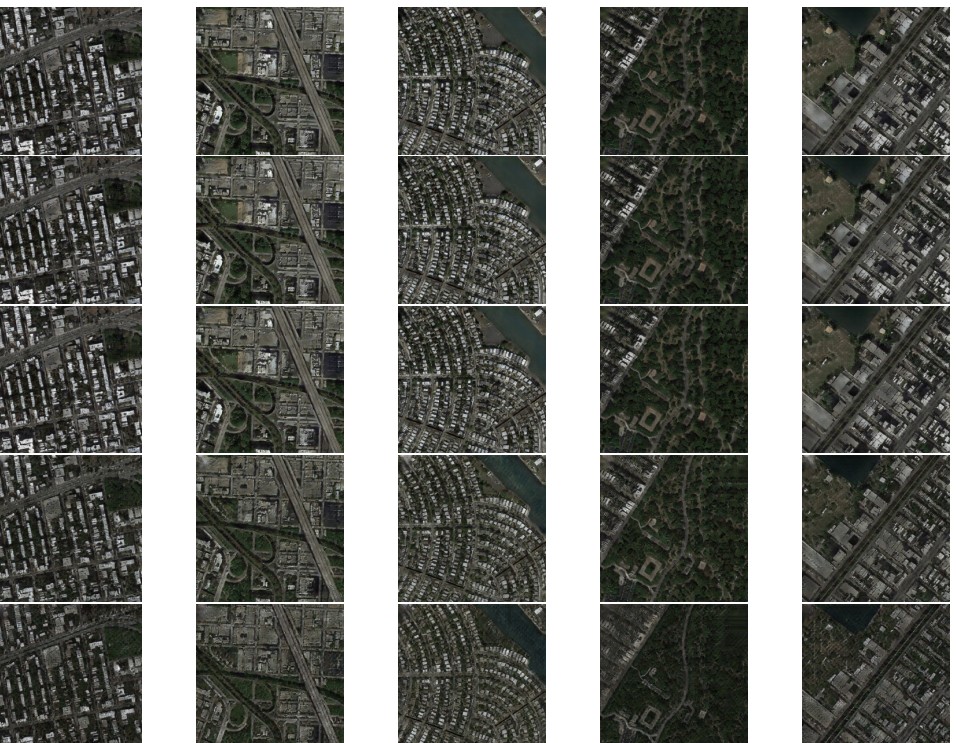

Figure 29: Domain translation: fine-grained pruning to different sparsity levels. Row 1: Output of the baseline generator. Rows 2-5: Synthesized satellite images by generators pruned to sparsities of 25%, 50%, 75%, 90%.

**CycleGAN**: photographic style applied to Monet's paintings.

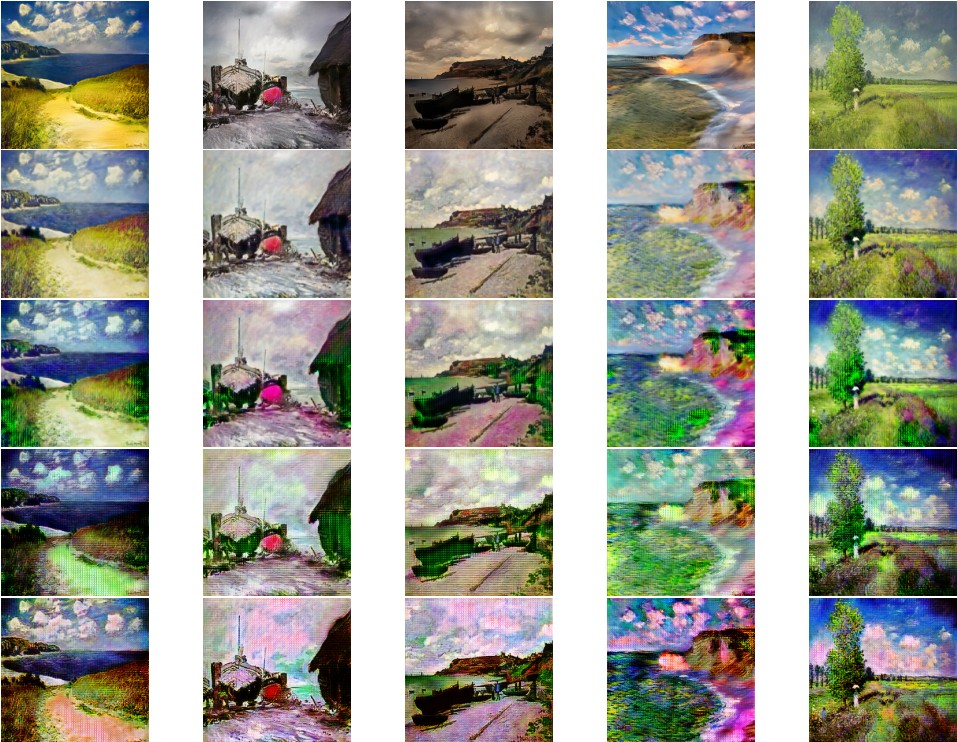

Figure 30: Style transfer: filter pruning to different sparsity levels. Row 1: Output of the baseline generator. Rows 2-5: Generated real photo style images by generators pruned to sparsities of 25%, 50%, 75%, 90%.

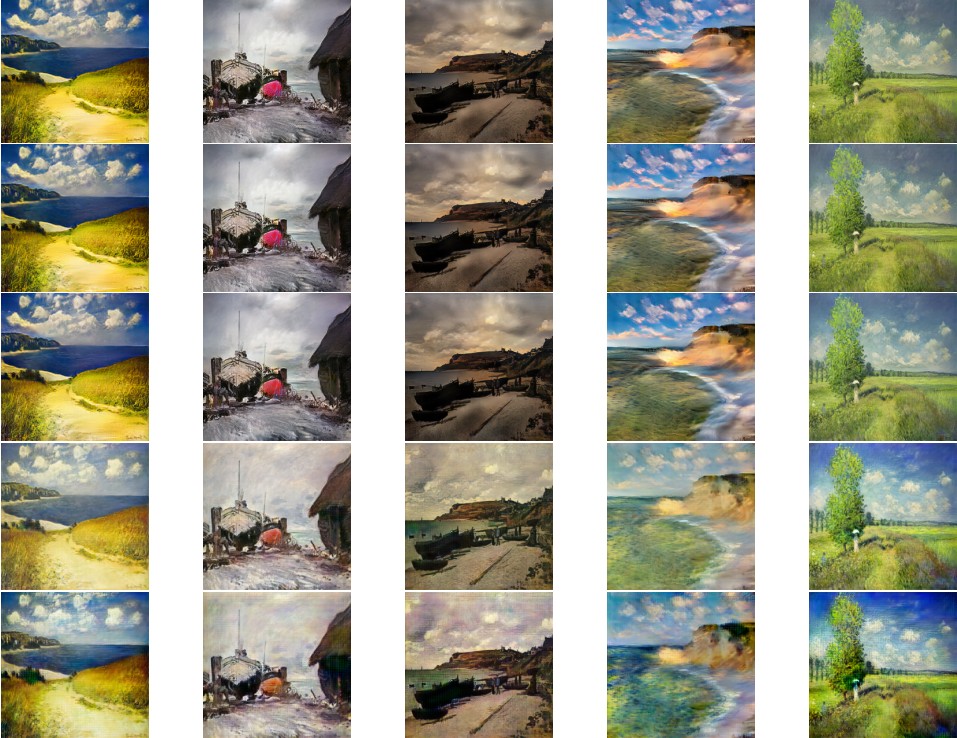

Figure 31: Style transfer: fine-grained pruning to different sparsities. Row 1: Output of the baseline generator. Rows 2-5: Generated real photo style images by generators pruned to sparsities of 25%, 50%, 75%, 90%.

**CycleGAN**: style of Monet's paintings applied to photographs.

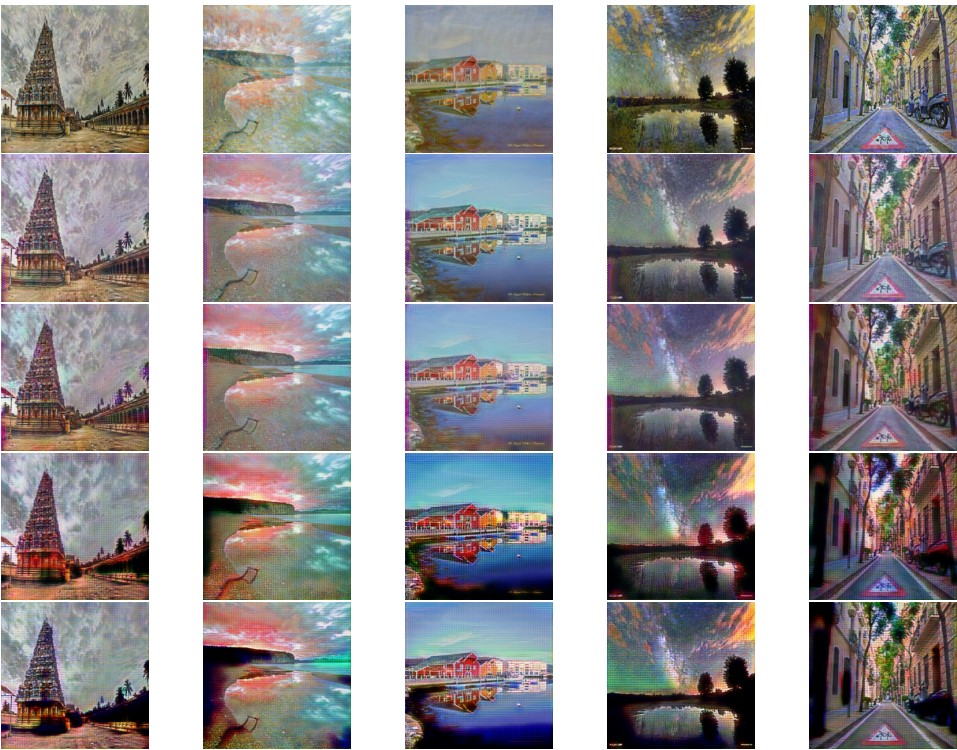

Figure 32: Style transfer: filter pruning to different sparsity levels. Row 1: Output of the baseline generator. Rows 2-5: Generated real photo style images by generators pruned to sparsities of 25%, 50%, 75%, 90%.

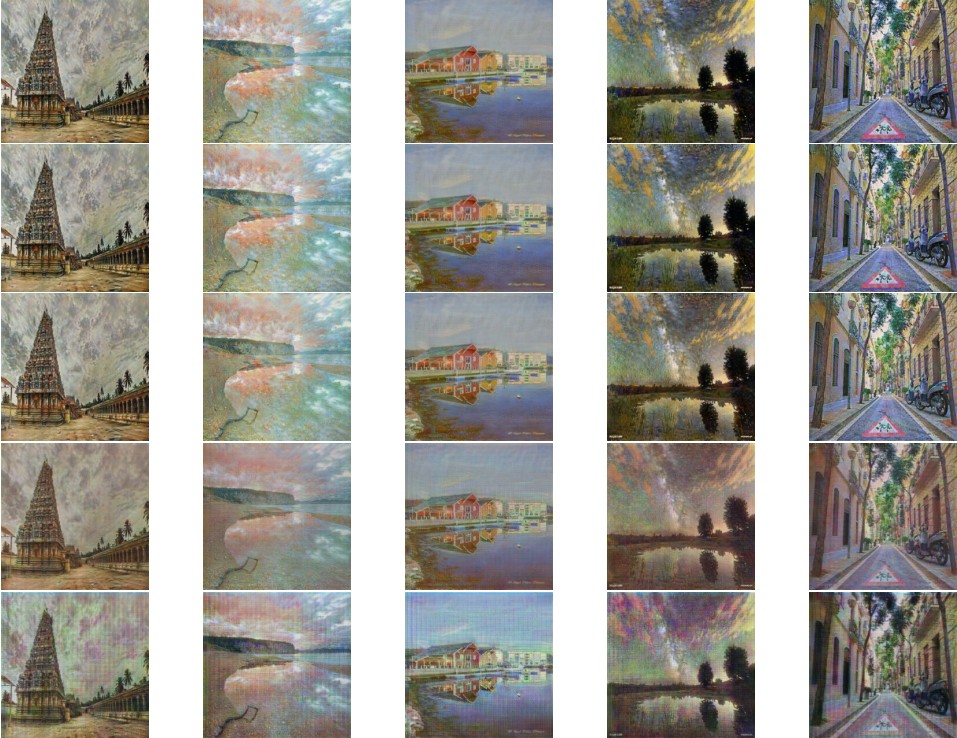

Figure 33: Style transfer: fine-grained pruning to different sparsities. Row 1: Output of the baseline generator. Rows 2-5: Generated real photo style images by generators pruned to sparsities of 25%, 50%, 75%, 90%.

**CycleGAN**: zebra to horse image-to-image translation.

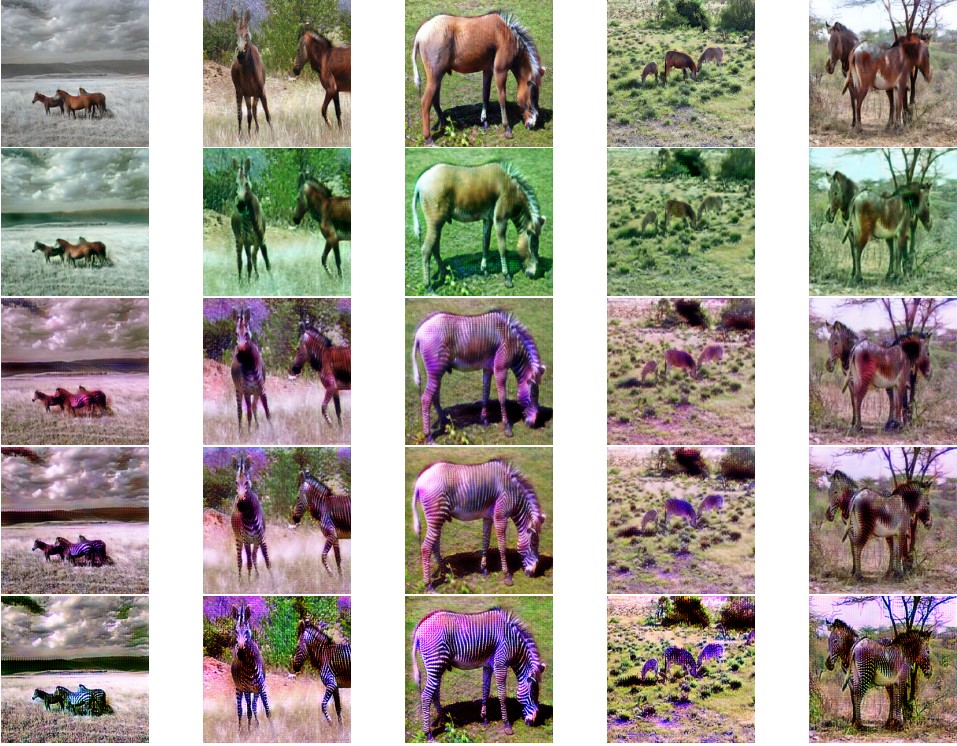

Figure 34: Image-to-image translation: filter pruning to different sparsities. Row 1: Baseline generator output. Rows 2-5: Generated real photo style images by generators pruned to sparsities of 25%, 50%, 75%, 90%.

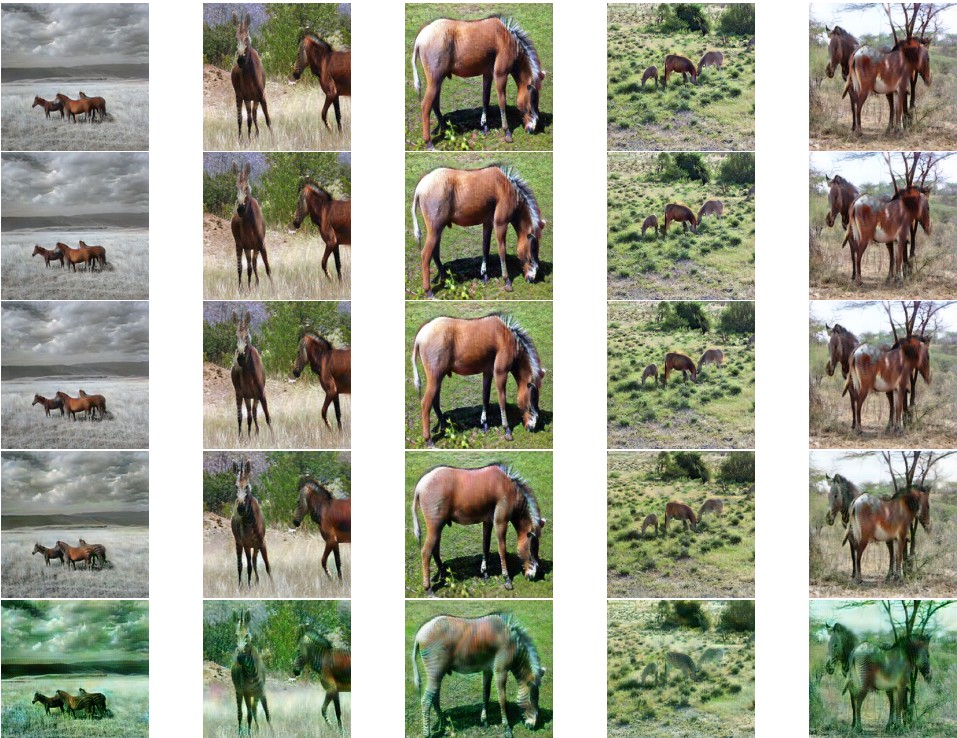

Figure 35: Image-to-image translation: fine-grained pruning to different sparsities. Row 1: Baseline generator output. Rows 2-5: Generated real photo style images by generators pruned to sparsities of 25%, 50%, 75%, 90%.

**CycleGAN**: horse to zebra image-to-image translation.

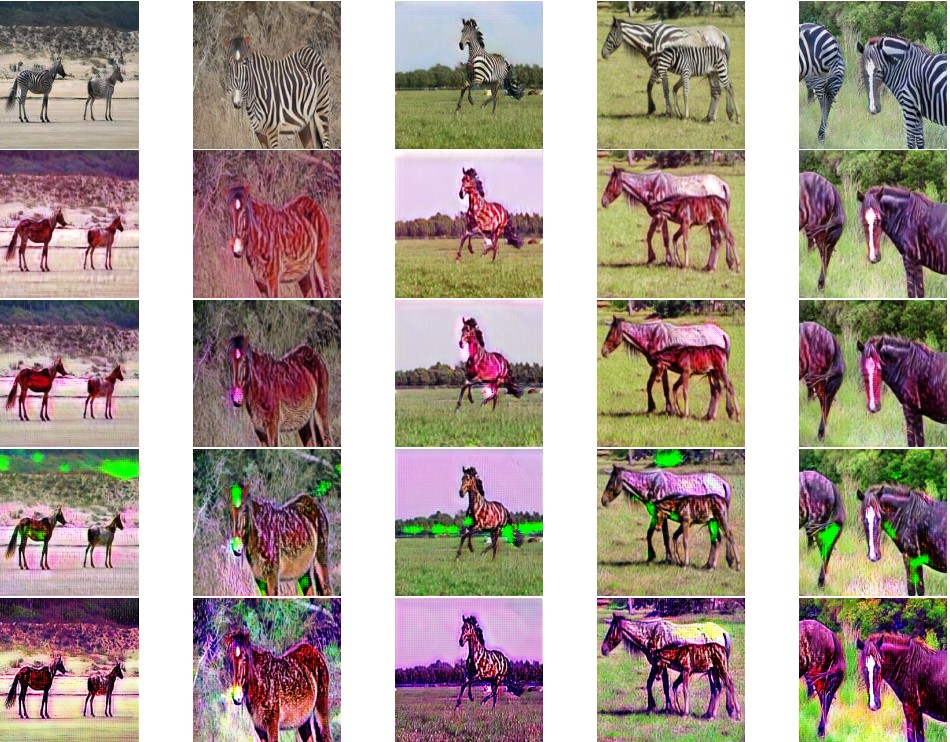

Figure 36: Image-to-image translation: filter pruning to different sparsities. Row 1: Baseline generator output. Rows 2-5: Generated real photo style images by generators pruned to sparsities of 25%, 50%, 75%, 90%.

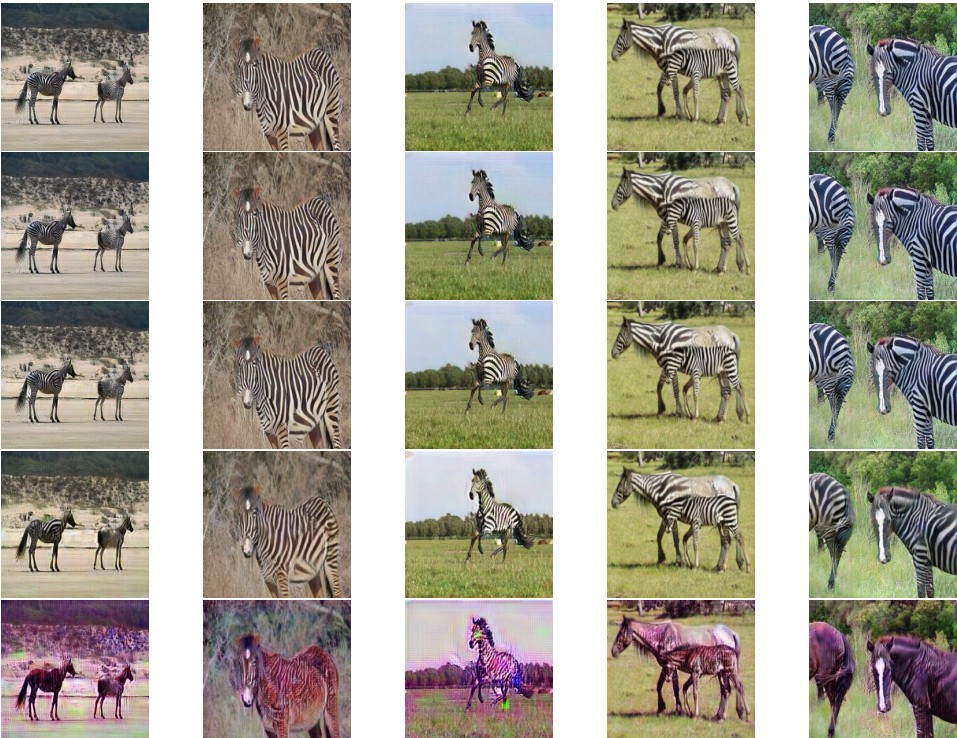

Figure 37: Image-to-image translation: fine-grained pruning to different sparsities. Row 1: Baseline generator output. Rows 2-5: Generated real photo style images by generators pruned to sparsities of 25%, 50%, 75%, 90%.

**StarGAN**: facial attribute image-to-image translation.

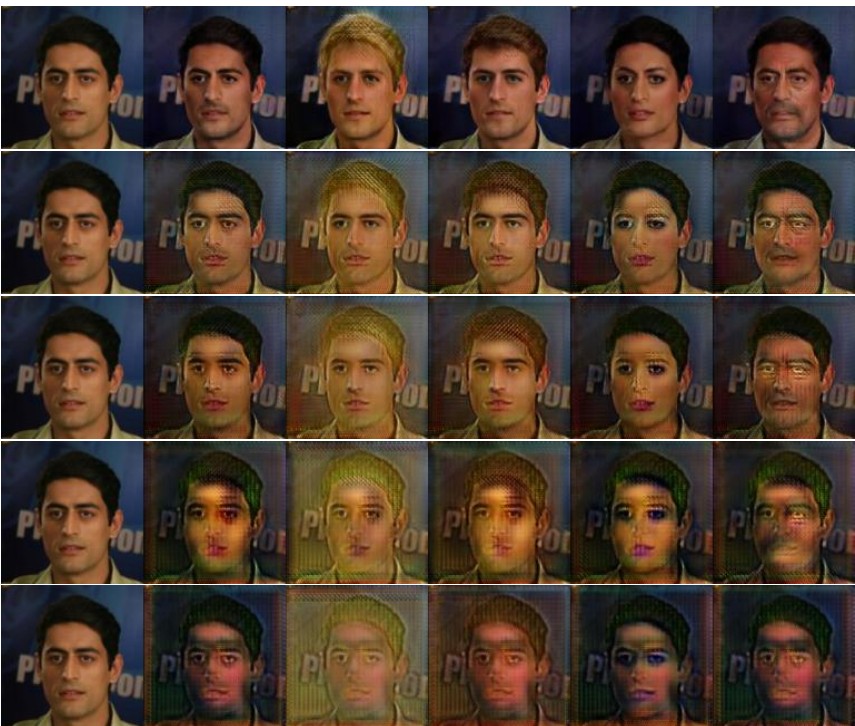

Figure 38: Image-to-image translation example 1: filter pruning to different sparsities. Row 1: Baseline generator output. Rows 2-5: Facial attribute translated images by generators pruned to sparsities of 25%, 50%, 75%, 90%.

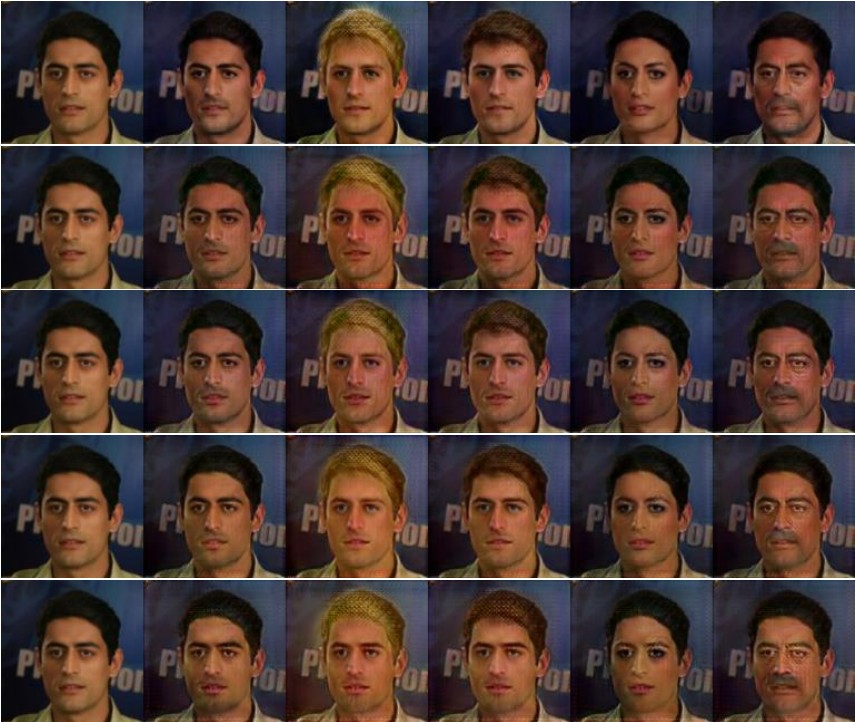

Figure 39: Image-to-image translation example 1: fine-grained pruning to different sparsities. Row 1: Baseline generator output. Rows 2-5: Facial attribute translated images by generators pruned to sparsities of 25%, 50%, 75%, 90%.

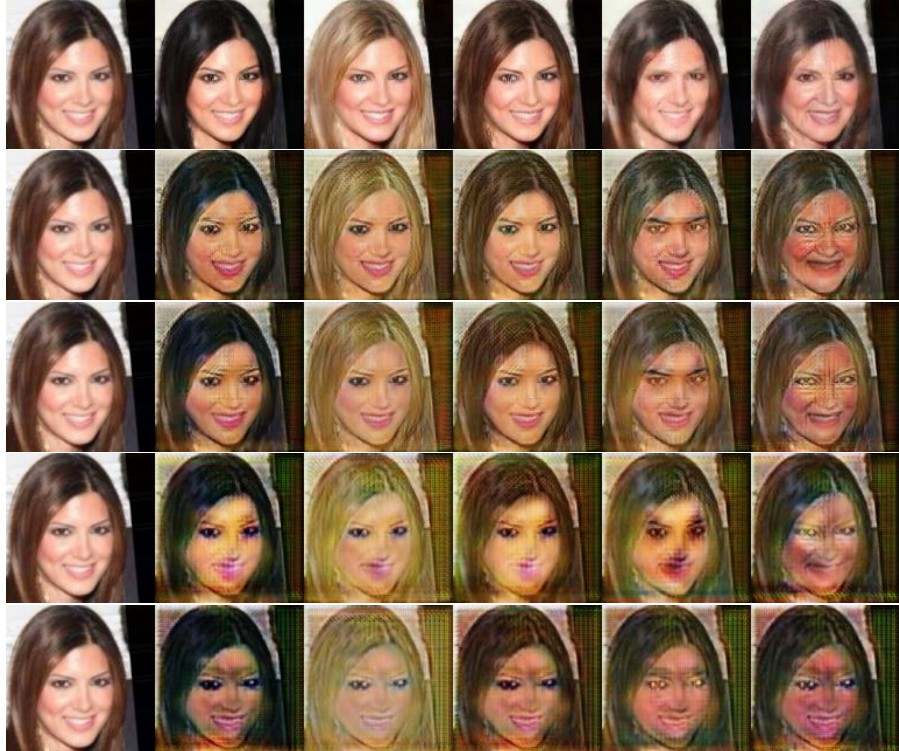

Figure 40: Image-to-image translation example 2: filter pruning to different sparsities. Row 1: Baseline generator output. Rows 2-5: Facial attribute translated images by generators pruned to sparsities of 25%, 50%, 75%, 90%.

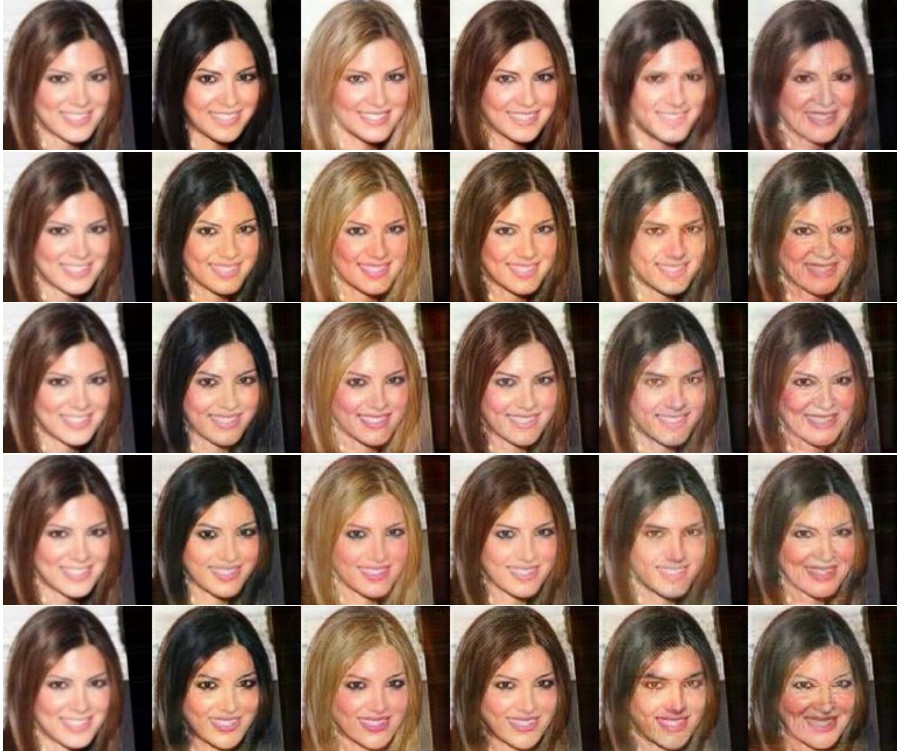

Figure 41: Image-to-image translation example 2: fine-grained pruning to different sparsities. Row 1: Baseline generator output. Rows 2-5: Facial attribute translated images by generators pruned to sparsities of 25%, 50%, 75%, 90%.

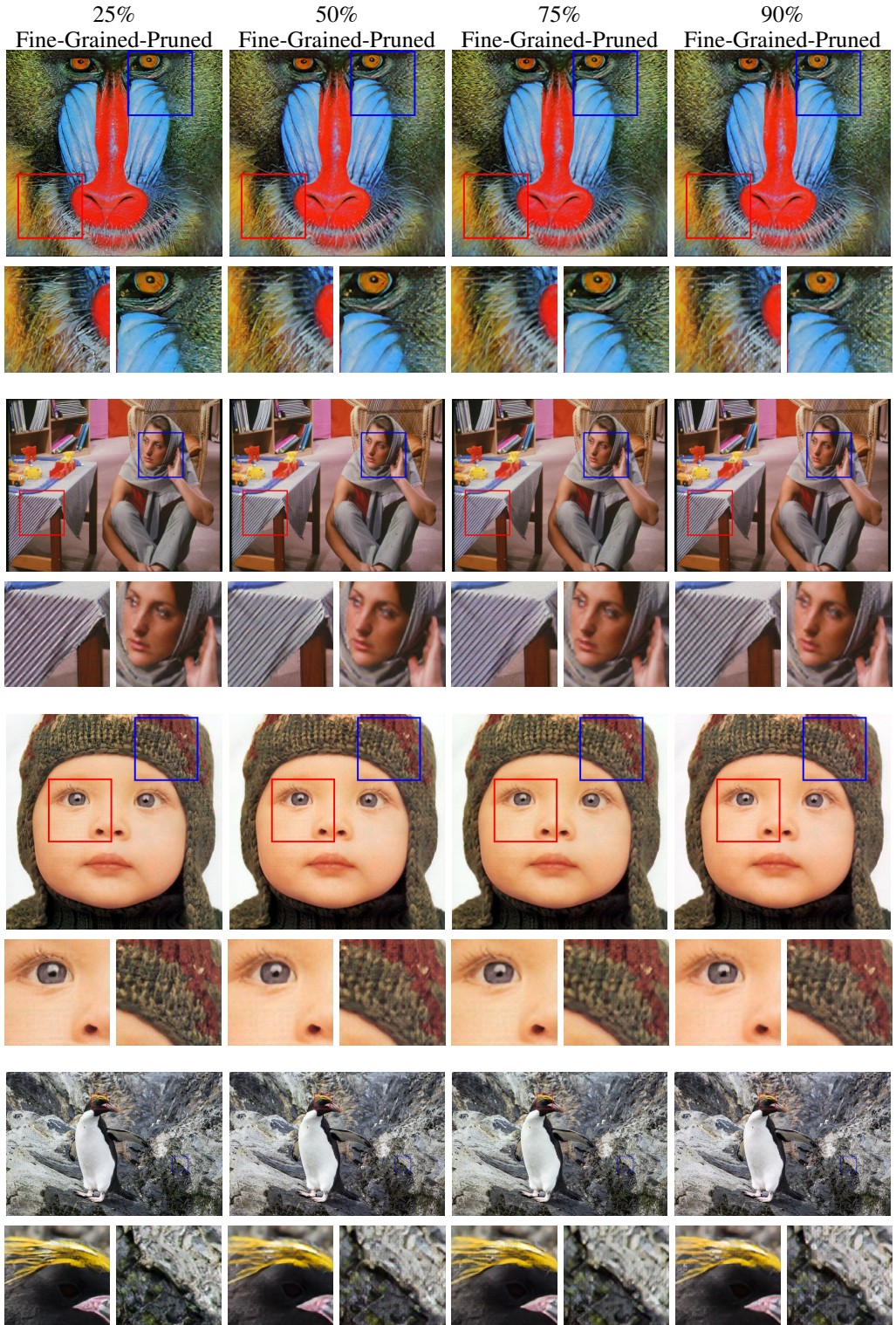

Figure 42: Super resolution: fine-grained pruning to different sparsities. Columns 1-4: Corresponding generated real high resolution images by generators pruned to sparsities of 25%, 50%, 75%, 90%.

The loss curves for the comparative experiment in Figure 38 and 40 are shown in Figure 43.

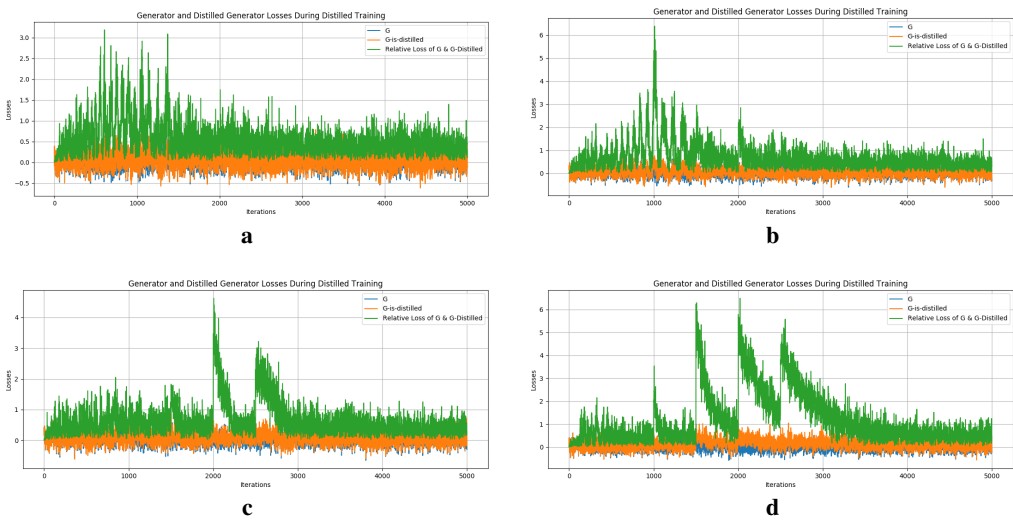

Figure 43: Loss curves of image-to-image translation experiments of filter pruning to different sparsities. **(a)**-**(d)**: Corresponding loss curve of the generator pruned to sparsities of 25%, 50%, 75%, 90%.

The loss curves for the comparative experiment in Figure 39 and 41 are shown in Figure 44.

Figure 44: Loss curves of image-to-image translation experiments of fine-grained pruning to different sparsities. **(a)**-**(d)**: Corresponding loss curve of the generator pruned to sparsities of 25%, 50%, 75%, 90%.

