# OpenReview forum: "Self-Supervised GAN Compression"
_ICLR.cc/2020/Conference — Reject_

### Official Review · AnonReviewer2 · 2019-10-16
**Official Blind Review #2**

**Rating:** 6

**Review:**

In this paper, the authors tackle the task of compressing a network. While there
are many effective solutions so far regular computer vision tasks, as they demonstrate,
they fail catastrophically  when applied to generative adversarial networks(GANs).
They propose a modification to the classic distillation method, where a
"student" network tries to imitate the uncompressed one under the supervision of
a fully converged discriminator network. They perform evaluation on multiple
tasks from image synthesis to super-resolution. They also study the influence
of the compression factor on the quality of the generated images.

The task is well motivated and situated in the related literature. The first
section is very thorough and extremely efficient at describing the failure modes
of existing methods. On one side, the results demonstrated in the evaluation are
compelling, on the other side, the compression factor is only 50%, which is much
lower than seen in related work. However, as it is shown in section 3 the task
may be much harder for GANS than regular models so I still consider it a
sizeable contribution.

There are a couple of points that require clarification. I personally found the
description of the method (Section 4) rather confusing. It is clear
what "discriminative loss" is as it is the one used in every GAN.
Unfortunately, I could not understand what "generative loss" means in the general
case. An example is given for StarGAN in equation (7) and I have a rough idea of
what to choose for Style Transfer, Domain Translation, Super Resolution and
Image translation. Though, it is unclear to me what to use in the case of
image synthesis. The experiments clearly show that it is possible so I think
it is necessary to show how this framework is concretely applied to each task at
hand.

During training, the discriminator only ever saw pictures from the true distribution
and the distribution generated by the generator (at each of its training steps).
If I understood the framework properly, here, the compressed generator is trained
from a random initialization. The distribution it outputs is therefore
completely unknown and potentially non overlapping with either of the true or
the generator ones. In that case it is hard to predict what the discriminator
would do on completely out of distribution samples. I seems reasonable to
conjecture that it might consider them "true" because it was never trained on
them. Could you provide an explanation of why it is not a problem in practice?
Do you have to try multiple initializations? Is the generative
loss enough to force the compressed discriminator to match the support
of the distribution of the dense generator?

I think this paper is novel, tackles a hard task and presents compelling results
(albeit using very mild compression ratios). It should be accepted if some
clarifications are made in section 3.

Minor Remarks:

- Figures 2, 9, 42 and 43 are unreadable when printed with a regular color office
  printer.
- It is unclear what it would take an extra 10% of the original number of epochs to train the compressed network. Why couldn't it be faster, or much longer?




**Experience Assessment:**

I have published one or two papers in this area.

**Review Assessment: Checking Correctness Of Derivations And Theory:**

N/A

**Review Assessment: Checking Correctness Of Experiments:**

I carefully checked the experiments.

**Review Assessment: Thoroughness In Paper Reading:**

I read the paper at least twice and used my best judgement in assessing the paper.

---

> ### Author Response · Authors · 2019-11-13
> **Response to Review #2**
>
> Thank you for your thorough review. We'd like to respond to some particular comments:
>
> >> Only 50% compression
>
> We chose to focus on 50% compression in early sections to highlight how easily existing methods fail at pruning generators and, in contrast, that ours succeeds. We share results for higher compression rates (up to 90%) using our method in Section 6, though we stress that we haven't spent any effort trying to find fancy fine-tuning schedules for each task, which may result in even higher compression rates.
>
> >> Generative loss
>
> Each task already has a generative loss; this is not a term that we have added. Rather, we modify the existing loss function to take into account the new (compressed) generator.  Here's what the TensorFlow DCGAN (image synthesis) tutorial has to say about the generator's loss term:
> "The generator's loss quantifies how well it was able to trick the discriminator. Intuitively, if the generator is performing well, the discriminator will classify the fake images as real (or 1). Here, we will compare the discriminators decisions on the generated images to an array of 1s.
>
> def generator_loss(fake_output):
>     return cross_entropy(tf.ones_like(fake_output), fake_output)"
> (https://www.tensorflow.org/tutorials/generative/dcgan)
> We simply add a new loss term, identical to this one, for the compressed generator. Since each model and task will have its own unique loss terms, we feel the best place to see these particular loss functions is in the baseline implementations.
>
> >> If I understood the framework properly, here, the compressed generator is trained from a random initialization.
>
> Your understanding is slightly incorrect – the compressed generator is initialized from the uncompressed generator (“Init Scheme” column in Table 1), which is trivial when the applied compression is something like pruning or quantization. In this way, the compressed sample distribution starts out close to the dense generator’s. We've added some clarification to Section 4. We have used different random seeds in some experiments with stable results.
>
> >> Figures 2, 9, 42 and 43 are unreadable when printed with a regular color office printer.
>
> We're working on Figure 2!  We've made it larger in the current revision.  The others, all in the appendix, may simply be best viewed in a digital format that supports zooming.  (As suggested by another reviewer, we may move the appendix material to a website.)
>
> >> Extra epochs
> 10% is simply what we found to be the maximum needed for good results; others took only an extra 1% of the original training epochs. This is a hyperparameter that one may choose to tune for more aggressive sparsities.

---

### Official Review · AnonReviewer1 · 2019-10-20
**Official Blind Review #1**

**Rating:** 6

**Review:**

This paper proposes a method to compress GANs. The motivation is that the current compression methods work for other kinds of neural networks (classification, detection), but perform poorly in the GAN scenario. The authors present intuitive reasons for why this is the case.

However, the motivation why we would like to compress GANs is unclear to us. The intro mentions: reducing memory requirements and improving their performance. Sure, compressing networks for object detection and classification on mobile devices is really useful. But GANs are mainly used for unsupervised density estimation, why put a GAN generator on a mobile device? But maybe we are missing something here.

Their “self-supervised” method works by using the pre-trained discriminator network, while compressing only the generator. They show both qualitative and quantitative gains.

The paper is clear and well-written. It presents a way of pruning GAN generator network and although of limited novelty, it might be an interesting read as it provides extensive and convincing experiments in a clear manner. It does have several parts though which require additional clarification.

The idea of using the pre-trained discriminator network seems reasonable, but I am missing what the compression method for the generator network actually is (Section 4). From Table 2 I would assume it is pruning, in which case the paper’s contribution is very limited.

The authors claim that the “self-supervised” method generalizes well to new tasks and models. "Generalizes" seems a strong word here, since the procedure compresses only the generator network. A more appropriate way of putting it might be ‘can be applied to other tasks and models.'

In Section 4 the authors write: “Our main insight is found,” but then they describe the GAN method. What is the actual insight there?

The qualitative results in Figure 1 suggest that their “self-supervised” method is better than the other baselines.

Scores from Table 2 also support the claims, but the table itself is not referenced anywhere in the text.

The analysis in Section 6 seems out of context with the rest of the paper. It is not clear how it relates to the “self-supervised” method.

Missing related work: 1st paragraph: compressing or distilling one network into another is much older than 2015, dating back to 1991 - see references in section 2 of the overview http://people.idsia.ch/~juergen/deep-learning-miraculous-year-1990-1991.html
The GAN principle itself is also much older (1990) - see references in section 5 of the link above.

General remarks:

In the first read of Section 3 it is not clear what [a], [b], [c] are.

It would be good to first refer to Table 1.

Table 1: why is there a “?” only on the “Fixed” column?

It would be good to have a larger font size in Figure 2, at least the size of the main text font.

In its current form, the pdf file has 100MBs (8MBs the main paper and the rest is the appendix). One could instead move the images from the appendix to a website and provide a link.

We might improve our rating provided the comments above were addressed in a satisfactory way in the rebuttal.



**Experience Assessment:**

I have published in this field for several years.

**Review Assessment: Checking Correctness Of Derivations And Theory:**

N/A

**Review Assessment: Checking Correctness Of Experiments:**

I assessed the sensibility of the experiments.

**Review Assessment: Thoroughness In Paper Reading:**

N/A

---

> ### Author Response · Authors · 2019-11-13
> **Response to Review #1**
>
> Thank you for your feedback and comments - these suggestions will make our submission stronger. Some responses to particular points follow:
>
> >> ... why put a GAN generator on a mobile device?
>
> Any real-time service using GANs, on a mobile device or otherwise, can benefit from model compression. General examples include mobile applications that perform style transfer, or video players that perform super-resolution on the client to save broadcast bandwidth. In the future, visual artists may rely on inpainting or other texture-generation techniques to save on asset storage space or interactive video generation to save rendering time, and musical artists may want a backing track to generate novel accompaniment that responds in real-time; all these tasks can be approached with GANs and may not work well with the latency associated with server-side execution. GANs have been used to augment training data, so, even in data center scenarios, having a more efficient generator can leave more resources available to training what may be a much more complex network.
>
> >> ... particular compression method ... pruning limits contribution
>
> Correct, we only present results for network pruning.  However, given the much better results with our method, we believe it may help other techniques achieve more aggressive compression rates. (We leave this as future work in our conclusion.)  Further, we have shown that network pruning fails spectacularly on generators in the absence of our technique, which is a surprising result we have not seen reported before.
>
> >> "Generalizes" --> "Applies to"
>
> We've addressed this to avoid making too broad a claim.
>
> >> In Section 4 the authors write: “Our main insight is found,” but then they describe the GAN method. What is the actual insight there?
>
> Our insight is, in fact, the next paragraph – we've fixed this to make it clear that reviewing the GAN method (the paragraph in question) leads to the insight (the next paragraph) of how to make fine-tuning a compressed model more stable and successful by using the (pre-trained) discriminator.
>
> >> Scores from Table 2 also support the claims, but the table itself is not referenced anywhere in the text.
>
> What an oversight!  We've fixed this.
>
> >> The analysis in Section 6 seems out of context with the rest of the paper. It is not clear how it relates to the “self-supervised” method.
>
> Weight pruning can be used to remove entire filters, not just individual elements (as noted by Reviewer #4), and 50% compression is somewhat modest (as noted by Reviewer #2). So, we included results for both filter pruning and pruning (elements and filters) more aggressively, to show that our method is successful, at least in more aggressive sparsity. While filter pruning was not as successful, this is yet more empirical evidence that pruning generators is not as straightforward a task as pruning classification or detection networks.
>
> >> Missing related work
>
> Thanks for pointing these out - we've added this extra background.
>
> >> It would be good to first refer to Table 1.
>
> Thank you for suggestion; this helps make the notation clear.
>
> >> Table 1: why is there a “?” only on the “Fixed” column?
>
> We've removed the ‘?’ from the “Compressed” and “Fixed” columns.
>
> Finally, we'll work on fixing the font size for Figure 2 and finding a better solution to the sizable appendix.

---

### Official Review · AnonReviewer4 · 2019-11-05
**Official Blind Review #4**

**Rating:** 3

**Review:**

The problem tackled in the paper is the compression of generators in adversarially trained models. Considering the success of large generative models (like BigGAN) one may wonder if these models can be compressed after being trained to improve practical applicability.

This paper is focused on the compression of image to image translation models and uses distillation on discriminator's outputs to achieve better results.

My decision is weak reject.

The message of the article is misleading. The whole goal of pruning a neural network is to remove filters from it, therefore, reducing the computation or, at least, storage space for the parameters. The attempt to remove filters was presented in the last figure, and it does not work as good as all other results presented in the paper.

The comparison in Figure 1 is arguably misleading as well. For example, one of the methods that were mentioned (LIT) does achieve a factor of 1.8 model compression, yet the comparison was not carried out directly with that method, but a modification proposed by the authors of this paper.

I would like to see more comparisons in terms of FLOPs or inference time between the baselines, SotA methods, and your proposed method. Weights pruning is simply one of the approaches for model compression, so you cannot ignore the alternatives.

Also, section 4 probably has to be rewritten, since some unorthodox notation is used. The authors should consider using some reference paper for the notations, like CycleGAN, that was mentioned in the paper. That will improve the clarity and readability of the used objectives.

**Experience Assessment:**

I have read many papers in this area.

**Review Assessment: Checking Correctness Of Derivations And Theory:**

I assessed the sensibility of the derivations and theory.

**Review Assessment: Checking Correctness Of Experiments:**

I assessed the sensibility of the experiments.

**Review Assessment: Thoroughness In Paper Reading:**

I read the paper at least twice and used my best judgement in assessing the paper.

---

> ### Author Response · Authors · 2019-11-13
> **Response to Review #4**
>
> Thank you for your time and feedback.  Please see our responses, below.
>
> >> Considering the success of large generative models (like BigGAN) one may wonder if these models can be compressed after being trained to improve practical applicability.
>
> This is a great question - we look forward to extending our technique to the state-of-the-art GANs available today. Looking at other domains, pruning has still been successful on larger, more capable networks.
>
> >> The message of the article is misleading. The whole goal of pruning a neural network is to remove filters from it
>
> We respectfully disagree with the assertion that the goal of pruning a network is to remove filters. This is one common approach used, but fine-grained (per-weight) pruning is not without merit - we show that our technique makes fine-grained pruning of generators possible; without this step, pruning filters would have no hope. Further, there are architectures (e.g. Cnvlutin2[1], SCNN[2]) that accelerate fine-grained pruning, as well as software approaches that achieve higher performance on more traditional architectures ([3],[4]). While our presented results for filter pruning show that it does not perform as well as on other tasks, we see this as an exciting area for future research. As with those other tasks for which filter pruning succeeded, work demonstrating the success of fine-grained pruning preceded filter pruning.
>
> [1] https://arxiv.org/abs/1705.00125
> [2] https://arxiv.org/abs/1708.04485
> [3] https://arxiv.org/abs/1802.10280
> [4] https://arxiv.org/abs/1804.10223 (not directly applicable to convolution-based GANs, but it shows that fine-grained pruning can offer a measurable benefit to real problems)
>
> >> The comparison in Figure 1 is arguably misleading as well. For example, one of the methods that were mentioned (LIT) does achieve a factor of 1.8 model compression, yet the comparison was not carried out directly with that method, but a modification proposed by the authors of this paper.
>
> We agree – LIT, as originally reported, does achieve a 1.8x compression rate (noted at the bottom of our original page 3). The results in Table 1 and Figure 1 are with past approaches to compressed training or fine-tuning applied to model pruning. Put another way: using distillation on intermediate representations (LIT) and removing layers and removing layers achieves a reported compression rate of 1.8x. When performing distillation on intermediate representations (LIT) and pruning the generator, the quality at a compression rate of 2x is quantitatively and qualitatively worse than the original generator. We've tried to make the context for the results we report in Figure 1 clear in the caption, as well as in the supporting text.
>
> >> FLOPs or inference time
>
> Performance will vary wildly based on particular architecture and hardware selected and is unfortunately out of the scope of this submission, which is hardware and architecture agnostic. Spending time aggressively pruning, optimizing for performance, and measuring against other baselines may be future work.
>
> >> Weights pruning is simply one of the approaches for model compression, so you cannot ignore the alternatives.
>
> We agree that weight pruning is one of many approaches to compression, but we also point out that many approaches are orthogonal: taking the example of LIT, from above, one could apply weight pruning on top of the shallower network that results from the original LIT application. Quantization, also successful at compressing GANs in the past, has seen success when applied with pruning in other domains. This large cross product of comparisons is also out of the scope of this submission. Finally, without our submission, there would be no possibility of combining pruning with other techniques, as pruning has not been shown to succeed on generators prior to our work.
>
> Had our investigation into pruning generators been straightforward ("We tried it, and it works"), then performance would have been the a primary focus. Instead, we focused on explaining why naive pruning fails and devising a robust method that is able to prune some tasks up to 90% sparsity.
>
> >> Section 4 unorthodox notation
>
> We added more general forms of equations 1,2,4, and 5 - thanks for the suggestions! Does this ease the understanding of section 4?

---

### Author Response · Authors · 2019-11-13
**Updated submission**

We've updated the submission to address questions and take advantage of suggestions offered by the reviewers.

---

### Decision · Program_Chairs · 2019-12-19

**Decision:**

Reject

**Comment:**

The paper develops a new method for pruning generators of GANs. It has received a mixed set of reviews. Basically, the reviewers agree that the problem is interesting and appreciate that the authors have tried some baseline approaches and verified/demonstrated that they do not work.

Where the reviewers diverge is on whether the authors have been successful with the new method. In the opinion of the first reviewer, there is little value in achieving low levels (e.g. 50%) of fine-grained sparsity, while the authors have not managed to achieve good performance with filter-level sparsity (as evidenced by Figure 7, Table 3 as well as figures in the appendices). The authors admit that the sparsity levels achieved with their approach cannot be turned into speed improvement without future work.

Furthermore, as pointed out by the first reviewer, the comparison with prior art, in particular with LIT method, which has been reported to successfully compress the same GAN, is missing and the results of LIT have been misrepresented. While the authors argue that their pruning is an "orthogonal technique", and can be applied on top of LIT, this is not verified in any way. In practice, combination of different compression techniques is known to be non-trivial, since they aim to explain the same types of redundancies.

Overall, while this paper comes close, the problems highlighted by the first reviewer have not been resolved convincingly enough for acceptance.